# Generalization in Neural Operator: Irregular Domains, Orthogonal Basis, and Super-Resolution

## Abstract

Neural operators (NOs) have become popular for learning partial differential equation (PDE) operators. As a mapping between infinite-dimensional function spaces, each layer of NO contains a kernel operator and a linear transform, followed by nonlinear activation. NO can accurately simulate the operator and conduct super-resolution, i.e., train and test on grids with different resolutions. Despite its success, NO's design of kernel operator, choice of grids, the capability of generalization and super-resolution, and applicability to general problems on irregular domains are poorly understood. To this end, we systematically analyze NOs from a unified perspective, considering the orthogonal bases in their kernel operators. This analysis facilitates a better understanding and enhancement of NOs in the following: (1) Generalization bounds of NOs, (2) Construction of NOs on arbitrary domains, (3) Enhancement of NOs' performance by designing proper orthogonal bases that align with the operator and domain, (4) Improvement of NOs' through the allocation of suitable grids, and (5) Investigation of super-resolution error. Our theory has multiple implications in practice: choosing the orthogonal basis and grid points to accelerate training, improving the generalization and super-resolution capabilities, and adapting NO to irregular domains. Corresponding experiments are conducted to verify our theory. Our paper provides a new perspective for studying NOs.

## 1 Introduction

Partial differential equation (PDE) operators are widespread in science and engineering. However, traditional numerical methods are known to be slow and ill-suited for high-dimensional problems. As a result, there has been a surge in the popularity of utilizing deep learning techniques for operator learning. Neural operators (NOs) [20, 19, 8, 21] are among the most important models. As a mapping between infinite-dimensional function spaces, each layer of NO contains a kernel operator and a linear transform to convert the input function, followed by nonlinear activation, conducted numerically based on the discretization of the input function on a grid. With appropriate kernels, e.g., shift-invariant kernels in Fourier NO (FNO) [20], facilitates the construction of complex operators. Stacking multiple NO layers further enhances the operator's complexity and demonstrates its universal approximation capabilities [15]. Moreover, empirical evidence reveals that NO exhibits fast convergence and excellent generalization, making it a practical choice. In addition to its operator fitting and generalization abilities, NO can also perform super-resolution tasks. This involves training the model on a low-resolution grid and accurately predicting outcomes on a high-resolution grid. This capability expands the utility of NO beyond precise operator fitting and generalization, showcasing its versatility and accuracy in super-resolution applications.

Insufficient understanding surrounds the design of kernel operators, choice of grid points, and NOs' capabilities, despite their notable features in generalization and super-resolution tasks. For

instance, while the Fourier basis-based FNO has been established, alternative approaches employing polynomial basis [21] and wavelet basis [8, 28] have been proposed to enhance NO's performance in handling operators related to non-periodic functions and multiscale functions, respectively. However, the underlying reasons for their effectiveness through basis changes remain elusive. Besides, the effectiveness of the super-resolution effect in NO is currently based on empirical observations, and the factors influencing its efficacy remain unknown. Furthermore, NO is commonly trained on a predefined uniform grid with a predetermined sparsity level. The impact of utilizing grids with varying sparsity levels or randomly sampled grid points on NO's performance remains unexplored.

On the other hand, the applicability of NOs to general problems on irregular domains poses a significant challenge, mainly because the kernel operators utilized in popular models [20, 21, 28] are typically defined on regular bounded domains. Extending NO, based on orthogonal bases, to encompass general irregular domains remains a formidable task. The difficulty lies in effectively incorporating the physical information of the domain into the design of NO's basis and grid.

In this paper, we provide a novel perspective for studying NOs by examining the role of orthogonal bases within their kernel operators. The kernel operators in NOs are constructed such that their eigenfunctions are predefined orthogonal bases, and eigenvalues are trainable parameters. This unified view enables the analysis of NOs in various aspects. Firstly, we establish generalization bounds for NOs, considering them mappings between infinite-dimensional function spaces. Moreover, by carefully designing orthogonal bases for the input domain and functions, NOs can be constructed on irregular domains, improving generalization. The impact of grid points on NO convergence and generalization is also investigated. Additionally, we analyze factors influencing the super-resolution error in NOs. Our theory carries practical implications, such as selecting appropriate orthogonal bases and grid points to accelerate convergence, enhance generalization and super-resolution abilities, and adapt NOs to irregular domains. Extensive experiments are conducted to validate our theory, which sheds new light on understanding NOs and improving their properties in practical applications.

## 2  Preliminary

### 2.1  Notation & Problem Definition

**Notation**. We use $\|\cdot\|_2$ to denote vector 2-norm or matrix spectral norm, while $\|\cdot\|_{l^2}$ and $\|\cdot\|_{L^2}$ are the norms in $l^2$ and $L^2$ spaces, respectively. We use $|\cdot|$ to denote the cardinality of a set. For a metric space $(S, \rho)$ and $T \subset S$ we say that $\hat{T} \subset S$ is an $\epsilon$-cover of $T$, if $\forall t \in T$, there $\exists \hat{t} \in \hat{T}$ such that $\rho(t, \hat{t}) \leq \epsilon$. The $\epsilon$-covering number of $T$ is defined as [13, 29]: $\mathcal{N}(\epsilon, T, \rho) = \min\{|\hat{T}| : \hat{T} \text{ is an } \epsilon - \text{cover of } T\}$.

**Problem Definition**. We consider the operator learning problem in [20] on the (possibly irregular or unbounded) domain $\Omega \subset \mathbb{R}^d$, with the input function space $\mathcal{A} = \mathcal{A}(\Omega)$ and output function space $\mathcal{H} = \mathcal{H}(\Omega)$, and the operator to be learned $\mathcal{G} : \mathcal{A} \to \mathcal{H}$. Adding a project can extend the model to different input and output domains. We are given training data $\{f_j, \mathcal{G}(f_j)\}_{j=1}^{N_{\text{train}}}$, where $f_j \sim \mu$ are $i.i.d.$ samples from an unknown distribution $\mu$ over the functions supported on $\mathcal{A}$. We aim to approximate $\mathcal{G}$ by a neural operator (NO) $\mathcal{G}_\theta$, which requires discretization. Thus, the input functions are represented by their pointwise values on a discrete grid $\{\vec{x}, f_j(\vec{x})\} = \{x_i, f_j(x_i)\}_{i=1}^{N_{\text{grid}}}$. Labels are also discretized on the same grid $\{\vec{x}, \mathcal{G}(f_j)(\vec{x})\} = \{x_i, \mathcal{G}(f_j)(x_i)\}_{i=1}^{N_{\text{grid}}}$, and train $\mathcal{G}_\theta$ via minimizing $\mathcal{L}_{\text{train}}(\theta) = \frac{1}{N_{\text{train}}} \sum_{j=1}^{N_{\text{train}}} \frac{1}{N_{\text{grid}}} \|\mathcal{G}(f_j)(\vec{x}) - \mathcal{G}_\theta(f_j)(\vec{x})\|_2^2$. The corresponding (regular) test loss on the same grid $\vec{x}$ is $\mathcal{L}_{\text{test-reg}}(\theta) = \mathbb{E}_{f \sim \mu} \frac{1}{N_{\text{grid}}} \|\mathcal{G}(f)(\vec{x}) - \mathcal{G}_\theta(f)(\vec{x})\|_2^2$. We also consider the super-resolution task, i.e., the model can make predictions on all $x \in \Omega$, which is approximated by taking another different and usually finer grid $\vec{x}_{\text{test}}$ as input and yields the output function pointwise values on the new grid. Note that the characteristic of NO is to take in an arbitrary grid and output the values of the target function on this grid, and the grid size can be arbitrary [19, 20]. This reflects that NO is a mapping between infinite-dimensional spaces. The super-resolution test loss on a different grid $\vec{x}_{\text{test}}$ is $\mathcal{L}_{\text{test-sr}}(\theta) = \mathbb{E}_{f \sim \mu} \|\mathcal{G}(f) - \mathcal{G}_\theta(f)\|_{L^2(\Omega)}^2 \approx \mathbb{E}_{f \sim \mu} \frac{1}{N_{\text{grid,test}}} \|\mathcal{G}(f)(\vec{x}_{\text{test}}) - \mathcal{G}_\theta(f)(\vec{x}_{\text{test}})\|_2^2$.

### 2.2  Understanding Neural Operator

Given an input function $f : \Omega \subset \mathbb{R}^d \to \mathbb{R}^h$ where $h$ is the hidden/output dim, and the complete orthogonal basis set $\{\phi_i\}_{i=0}^\infty$ on $L^2(\Omega; \mathbb{R})$ arranged with increasing frequencies/degrees/orders, each

87  layer of NOs contains three operations: (1) the kernel transform, (2) the linear transform, and (3) the
88  nonlinear activation. The overall model structure can be summarized as follows.

89  **Overall Model**. Denote the input function as $\hat{u}_0(\boldsymbol{x})$, then the recursive formulation of NO is

$$\hat{u}_l(\boldsymbol{x}) = \sigma\left(u_l(\boldsymbol{x})\right),\ l \geq 1;\ \hat{u}_0(\boldsymbol{x}) = \hat{u}_0(\boldsymbol{x}),$$
$$u_{l+1}(\boldsymbol{x}) = \int K(\boldsymbol{B}_l, \boldsymbol{x}, \boldsymbol{y})\hat{u}_l(\boldsymbol{y})d\boldsymbol{y} + \boldsymbol{W}_l\hat{u}_l(\boldsymbol{x}), \tag{1}$$
$$v(x) = u_L(x),$$

90  where $v(x)$ is the output, $l$ is the layer index, and $L$ is the total number of layers. $\hat{u}$ and $u$ are the
91  post-activation and pre-activation input functions, respectively.

92  **Kernel Transform** $\kappa$. We set up trainable kernels such that the basis set with $N_{\text{modes}}$ lowest
93  frequencies form the eigenfunctions of the kernels, i.e., $K(\boldsymbol{B}; \boldsymbol{x}, \boldsymbol{y}) = \sum_{i=0}^{N_{\text{modes}}} \boldsymbol{B}_i\phi_i(\boldsymbol{x})\phi_i(\boldsymbol{y})$ where
94  $\boldsymbol{B}_i \in \mathbb{R}^{h \times h}$ and $\boldsymbol{B} \in \mathbb{R}^{N_{\text{modes}} \times h \times h}$ are trainable, and $h$ is the hidden dim. Thanks to the low-
95  rank property of the kernel, i.e., the dimension of its kernel spectra are only $N_{\text{modes}}$, $\kappa(f)(\boldsymbol{x}) =$
96  $\int K(\boldsymbol{B}; \boldsymbol{x}, \boldsymbol{y})f(\boldsymbol{y})d\boldsymbol{y} = \sum_{i=0}^{N_{\text{modes}}} \boldsymbol{B}_i\boldsymbol{c}_i\phi_i(\boldsymbol{x}) \in \mathbb{R}^h$, where $\boldsymbol{c}_i = \langle f, \phi_i \rangle \in \mathbb{R}^h$ is the dimension-wise
97  inner product between the input $f$ and the basis $\phi_i$. This form is exactly the same as the kernel in
98  FNO if the basis $\phi_i$ is Fourier series. For implementation, we usually (1) project the input function
99  onto the basis with $N_{\text{modes}}$ lowest frequencies to get $\{\boldsymbol{c}_i\}_{i=1}^{N_{\text{modes}}}$; (2) linear transform the coefficients
100 $\boldsymbol{c}_i$ with the matrices $\boldsymbol{B}_i$; (3) finally project from the coefficient space back to the function space by
101 multiplying $\phi_i(\boldsymbol{x})$ to each $\boldsymbol{B}_i\boldsymbol{c}_i$.

102 **Linear Transform** $\omega$. The linear transform $\omega$ is a straightforward operation $\omega[f](\boldsymbol{x}) =$
103 $\boldsymbol{W}f(\boldsymbol{x}), \boldsymbol{W} \in \mathbb{R}^{h \times h}, f(\boldsymbol{x}) \in \mathbb{R}^h$. Unlike the kernel operator, the linear transform can be con-
104 ducted in the original function space without projection. The final output is obtained by combining
105 the results from the kernel transform and the linear transform: $T[f](\boldsymbol{x}) = \kappa(f)(\boldsymbol{x}) + \omega[f](\boldsymbol{x})$.
106 Finally, the transformer function is passed through a nonlinear activation, i.e., $\sigma\left(T[f](\boldsymbol{x})\right)$.

107 **Examples of Bases**. In FNO [20], the orthogonal basis used is the Fourier basis, specifically over a
108 bounded regular domain. However, alternative orthogonal bases are employed in other variations
109 of FNO, such as those discussed in [8, 21], including polynomial and wavelet bases. Regarding the
110 implementation of the most popular FNO [20]: (1) is the fast Fourier transform (FFT); (2) is its
111 coefficient transform; (3) is the inverse FFT.

112 **Numerical Integration**. In practice, input functions are discretized on a grid. Transformations and
113 activations can be applied pointwise, but kernel transforms relying on the inner product with bases
114 require numerical integration on the grid. However, the linear transform does not need numerical
115 integration and can be implemented more easily in the original function space.

116 **Efficiency of Numerical Integration**. In FNO, the Fast Fourier Transform (FFT) is used to compute
117 integrals, with a time complexity of $O(N_{\text{grid}} \log N_{\text{grid}})$. However, in practice, only the first $N_{\text{modes}}$
118 integrations between the basis and input need to be calculated. This reduces the complexity to $O(N_{\text{grid}})$
119 since $N_{\text{modes}} \ll N_{\text{grid}}$. This approach, adopted in Geo-FNO [19], ensures that the integration step
120 does not become a bottleneck in the NO model's time complexity. As a result, NO models are
121 generally faster than Transformers.

## 3  Theory

123 We have examined the traditional function perspective of NOs. However, machine learning often
124 overlooks the complexity of mappings between infinite-dimensional functions. To address this, we
125 propose studying NOs in the coefficient space. By representing NOs as mappings between infinite
126 sequences of real numbers, derived from the expansion of input functions on an orthogonal basis, we
127 can leverage the extensive literature on mappings between finite-dimensional vectors and extend it to
128 our context. This enables a comprehensive analysis of NOs from a new perspective.

129 Given a complete orthogonal basis $\{\phi_i\}_{i=0}^{\infty}$, the input function $f$ and the output function $\mathcal{G}(f)$ can be
130 expanded as $f = \sum_{i=0}^{\infty} c_i\phi_i, \mathcal{G}(f) = \sum_{i=0}^{\infty} d_i\phi_i$ where $c_i = \langle f, \phi_i \rangle$ and $d_i = \langle \mathcal{G}(f), \phi_i \rangle$. For the
131 infinite sums to converge, the infinite sequences $\{c_i\}, \{d_i\} \in l^2$. So, the operator learning problem
132 on $\mathcal{G}$ can be abstracted to a mapping between infinite sequences between numbers in the $l^2$ space, i.e.,

NOs aim to learn the mapping $\{c_i\}_{i=0}^\infty \mapsto \{d_i\}_{i=0}^\infty$. Thus, we can define the input space from the viewpoint of coefficients $\mathcal{B} := \{(\langle f, \phi_i \rangle)_{i=1}^\infty, f \in \mathcal{A}\}$. where $\mathcal{A}$ is the space of input functions.

We reinterpreted the operations in NOs from the sequences mapping perspective in $l^2$.

**Kernel Transform**. It maps $f = \sum_{i=0}^\infty c_i \phi_i$ to $\kappa(f) = \int K(\boldsymbol{B}; x, y) f(y) dy = \sum_{i=0}^{N_{\text{modes}}} \boldsymbol{B}_i c_i \phi_i$, which can be abstracted to: $c_{\leq N_{\text{modes}}} \mapsto \boldsymbol{B}_{\leq N_{\text{modes}}} c_{\leq N_{\text{modes}}}, c_{>N_{\text{modes}}} \mapsto 0$, due to the kernel's local rankness and truncation at the $\overline{N}_{\text{modes}}$-lowest frequency.

**Linear Transform**. For an input function $f(x)$, the linear transform maps $f(x)$ to $\boldsymbol{W}f(x)$. Also, considering the channel-wise operation, from the sequence space point of view, it is $c_i \mapsto \boldsymbol{W}c_i$. Compared to the previous kernel transform, the linear coefficient is the same for all elements in the sequence, but those are different and truncated in the previous kernel transform.

**Nonlinear Activation**. The nonlinear activation $\sigma$ is an abstract and fixed mapping between $l^2$ to itself, denoted $\Sigma : l^2 \to l^2, c \mapsto \Sigma c$. In the original function space, the mapping is from the input function $f(x)$ to $\sigma(f)(x)$ where $\sigma : C(\Omega) \to C(\Omega)$. However, in the sequence space, the original $c_i$ is $c_i = \langle f, \phi_i \rangle$, and $(\Sigma c)_i = \langle \sigma(f), \phi_i \rangle$. The $i$th entry of $\Sigma c$ may depend on all $c_i$ for every $i$ since the activation is performed on the whole input function. For instance, for the input function $f(x) = \cos(kx)$ and the cosine basis, only $c_k = 0$ while other entries equal zero. But $\sigma(f)(x) = \sigma(\cos(kx))$ is a very complicated function even for simple $\sigma$ like ReLU, Sigmoid, and Tanh activations, with the post-transformed coefficients $(\Sigma c)_i \neq 0$ for all $i$. Although it is abstract, the Lipschitz continuity is kept:

**Proposition 3.1.** *If $\sigma$ is L-Lipschitz, i.e., $|\sigma(x) - \sigma(y)| \leq L|x - y|$, then the mapping $\Sigma$ is also L-Lipschitz in the $l^2$ space.*

**Model Summary**. Denote the input function as $\hat{u}_0(\boldsymbol{x})$, and its expansion over the orthogonal basis to be $\{\hat{c}_{0,i}\}_{i=1}^\infty$, i.e., $\hat{u}_0 = \sum \hat{c}_{0,i} \phi_i$, then the recursive formulation of NO in the coefficient space is

$$\hat{c}_{l,i} = \Sigma c_{l,i}, \quad l \geq 1;$$
$$c_{l+1,\leq N_{\text{modes}}} = (\boldsymbol{B}_{l,\leq N_{\text{modes}}} + \boldsymbol{W}_l)\hat{c}_{l,\leq N_{\text{modes}}}, \quad c_{l+1,>N_{\text{modes}}} = \boldsymbol{W}_l \hat{c}_{l,>N_{\text{modes}}}; \qquad (2)$$
$$v_i = c_{L,i};$$

where $v_i$ is the coefficient for the output function, i.e., the output function $v(\boldsymbol{x}) = \sum_{i=1}^\infty v_i \phi_i(\boldsymbol{x})$, $l$ is the layer index, and $L$ is the total number of layers. For all the indices $c_{l,i}$, the first $l$ is for the layer, while the second $i$ is for the index of the orthogonal basis. $\hat{c}$ and $c$ are the post-activation and pre-activation input coefficients, respectively.

## 3.1 Generalization of NOs

From the sequence perspective, we can derive the generalization bound of NOs via the robustness bound [29, 13]. The generalization gap of the model given in equation (2) can be bounded as follows.

**Theorem 3.1.** *(Generalization bound of NOs) For any $\delta \in (0, 1)$, with probability at least $1 - \delta$ over the choice of random samples $S = \{f_j\}_{j=1}^{N_{\text{train}}} \sim \mu$, let the model parameters after optimization to be $\theta_S = \{\{\boldsymbol{B}_{l,i}\}_{i=1}^{N_{\text{modes}}}, \boldsymbol{W}_l\}_{l=0}^L$, the following bound holds:*

$$|L_{\text{test-reg}}(\theta_S) - L_{\text{train}}(\theta_S)| \leq \prod_{l=0}^L \left( \max_i \{\|\boldsymbol{B}_{l,i} + \boldsymbol{W}_l\|_2, \|\boldsymbol{W}_l\|_2\} \right) \gamma + M \sqrt{\frac{2K \log 2 + 2 \log(1/\delta)}{N_{\text{train}}}}, \quad (3)$$

*for all $\gamma > 0$, where $K = \mathcal{N}(\gamma/2, \mathcal{B}, \|\cdot\|_{l^2})$ is the $\gamma/2$-covering number of the input space $\mathcal{B}$ under the norm $\|\cdot\|_{l^2}$. $M$ is the upper bound of the loss function.*

All proofs are presented in the Appendix. The generalization bounds, similar to vanilla neural nets, rely on the products of multilayer parameter norms [2, 29]. Theorem 3.1 offers a more detailed characterization of the generalization bounds compared to the findings in [14], and it guides selecting orthogonal bases in NO. We will delve into this topic further in Section 4.

**Extension to Discretized NOs**. In Theorem 3.1, we primarily focus on continuous NOs. However, the presented theory can be extended to discrete NOs by substituting the infinite-dimensional $l^2$ space with a finite-dimensional vector space. In this context, inner products and orthogonal bases can still be defined. The modification lies in the term involving the covering number in the bound $\mathcal{N}(\gamma/2, \mathcal{B}, \|\cdot\|_{l^2})$. Here, we replace the $l^2$ space and norm with the finite-dimensional Euclidean space corresponding to the discrete NO and its vector 2-norm.

## 3.2 Super-resolution Error

Super-resolution involves training a model on a low-resolution grid and evaluating it on a high-resolution grid, with the expectation of comparable performance. While FNO [20] demonstrates excellent super-resolution capabilities, the underlying reasons remain poorly understood. This understanding is crucial for two reasons: (1) enabling training on sparse grids, leading to reduced training time, and (2) ensuring NOs can effectively handle inputs of the same function on different grids and produce satisfactory results.

Before delving into the analysis, it is important to address the numerical integration errors that arise when training NOs on low-resolution grids compared to high-resolution grids during super-resolution. Intuitively, if the integration error is significant, there will be notable discrepancies in the integral values obtained from the sparse training grid and the high-resolution testing grid, resulting in inconsistent model performance between training and testing.

As $u_l$, $\hat{u}_l$, and $v$ represent variables in continuous NOs, we use $U_l$, $\hat{U}_l$, and $V$ to represent variables in the discrete NOs using numerical integration rule $\hat{\int} g \approx \sum_{i=1}^{N_{\text{grid}}} w_i g(\boldsymbol{x}_i)$ for any integrand $g$:

$$\hat{U}_l(\boldsymbol{x}) = \sigma\left(U_l(\boldsymbol{x})\right), \ l \geq 1; \quad V = U_L; \quad U_0 = f;$$

$$U_{l+1}(\boldsymbol{x}) = \hat{\int} K(\boldsymbol{B}_l, \boldsymbol{x}, \boldsymbol{y})\hat{U}_l(\boldsymbol{y})d\boldsymbol{y} + \boldsymbol{W}_l\hat{U}_l(\boldsymbol{x}) = \sum_{i=1}^{N_{\text{grid}}} w_i K(\boldsymbol{B}_l, \boldsymbol{x}, \boldsymbol{x}_i)\hat{U}_l(\boldsymbol{x}_i) + \boldsymbol{W}_l\hat{U}_l(\boldsymbol{x}),$$

where $f$ is the input function, $w_i$ is the weight for numerical integral at the grid point $\boldsymbol{x}_i$. The error in numerical integration generally depends on two factors: the grid size (defined as $e_{\text{grid}}(N_{\text{grid}})$) and the smoothness of the integrand function (defined as $e_{\text{func}}(f)$).

For instance, on a uniform grid over the interval $[a, b]$, the integral is approximated using the Darboux method $\int_a^b f \approx 1/N_{\text{grid}} \sum_{i=1}^{N_{\text{grid}}} f(x_i)$ where $x_i = a + (i-1)(b-a)/N_{\text{grid}}$, which yields an integration error of $O(f''(\xi)/N_{\text{grid}})$ where $\xi \in (a, b)$, i.e., $e_{\text{grid}}(N_{\text{grid}}) = 1/N_{\text{grid}}^2$, $e_{\text{func}}(f) = f''(\xi)$. However, using the trapezoidal rule instead, the error can be reduced to $O(f''(\xi)/N_{\text{grid}}^2)$ without requiring additional computations. FNO [20] assumes that the input function is periodic, so the error of the uniform grid decreases to $O(f''(\xi)/N_{\text{grid}}^2)$. For the Gaussian quadrature on the interval $[a, b]$, the error can be reduced to $O\left((N_{\text{grid}}!)^4 f^{2(N_{\text{grid}})}(\xi)/[(2N_{\text{grid}})!]^3\right)$ for $\xi \in (a, b)$ [10]. With the background, we can write the discretization error of NOs:

**Theorem 3.2.** *(Discretization error of NOs) Suppose the numerical integration's error is $e_{grid}(N_{grid})e_{func}(f)$ where $N_{grid}$ is the grid size, and $f$ is the integrand, then the discretization error of discrete NOs compared with continuous ones due to numerical integral is upper bounded by*

$$\|v - V\|_{L^2} \leq \sum_{l=0}^{L} \prod_{k=l}^{L} \max_i \left\{\|\boldsymbol{B}_{k,i} + \boldsymbol{W}_k\|_2, \|\boldsymbol{W}_k\|_2\right\} \left(\sum_{i=0}^{N_{modes}} \|\boldsymbol{B}_{i,l}\|_2 e_{grid}(N_{grid})e_{func}\left(\hat{U}_l \cdot \phi_i\right)\right). \quad (4)$$

The discretization error in discrete NOs relies on the norm of model parameters and the accuracy of the integration method employed for integrating intermediate output functions. Building upon this, we can derive the super-resolution error of discrete NOs. Firstly, we bound the prediction error of continuous NOs across all points in the domain $\Omega$ (i.e., the super-resolution error of continuous NOs). During the training phase, NOs are trained on a finite training grid, leading to this error. Subsequently, we bound the discrepancy between continuous and discrete NOs, corresponding to the discretization error stated in Theorem 3.2. These two terms are reflected in the following theorem.

**Theorem 3.3.** *(Super-resolution error of NOs) Assuming a uniform grid on a bounded regular domain with $N_{grid}$ points in the FNO [20] setting, under the same notation as Theorem 3.1. Then, the super-resolution error of the discrete NO model for the input function $f$ with intermediate output $\hat{U}_l$ (as detailed in equation (3.2)) can be bounded as follows:*

$$|L_{test\text{-}sr}(\theta_S) - L_{test\text{-}reg}(\theta_S)| \leq \sum_{l=0}^{L} \prod_{k=l}^{L} \max_i \left\{\|\boldsymbol{B}_{k,i} + \boldsymbol{W}_k\|_2, \|\boldsymbol{W}_k\|_2\right\} \left(\sum_{i=0}^{N_{modes}} \|\boldsymbol{B}_{l,i}\|_2 e_{grid}(N_{grid})e_{func}\left(\hat{U}_l \cdot \phi_i\right)\right)$$

$$+ \left\{\sum_{i=0}^{N_{modes}} \sum_{l=0}^{L-1} \left(\prod_{k=l+1}^{L-1} \|\boldsymbol{W}_k\|_2\right) \|\boldsymbol{B}_{l,i}\langle \hat{u}_l, \phi_i\rangle\|_2 Lip(\phi_i) + \prod_{l=0}^{L-1} \|\boldsymbol{W}_l\|_2 Lip(f)\right\}/N_{grid}.$$

$$(5)$$

The first term pertains to the integral error in Theorem 3.2, while the second term relates to the interpolation and generalization ability of NOs across the entire domain. These factors significantly influence the super-resolution of NOs. Understanding and addressing these factors is crucial for enhancing super-resolution accuracy in NOs, which will be thoroughly discussed in Section 4.

## 4 Implication and Application of the Theory

In this subsection, we introduce the implications and applications of the proposed theory and correspond them to the following numerical experiments.

**Tighter Bound in Theorem 3.1**. Our bound's proof and form are much more general and tighter than previous work [14]. In particular, in terms of parameter matrix norm contributed by each layer, the bound in [14] depends on $\|\boldsymbol{W}_l\|_F + \|\boldsymbol{B}_l\|_F N_{\text{modes}}^{d/2}$ where $\|\cdot\|$ denotes the Frobienus norm. So, our reliance of $\max_i \{\|\boldsymbol{B}_{l,i} + \boldsymbol{W}_l\|_2, \|\boldsymbol{W}_l\|_2\}$ is a significant improvement. Additionally, our bound suggests that increasing the number of modes does not necessarily increase the complexity of the model. However, it is important to note that selecting high-frequency basis functions to fit high-frequency noise can adversely impact generalization, as it leads to larger values of $\|\boldsymbol{B}_{l,i} + \boldsymbol{W}_l\|_2$. We shall verify the advantage of our bound in Experiment 6.1.

**Super-Resolution Error**. From Theorem 3.3, super-resolution in NOs depends on two critical factors: (1) the accuracy of integration trained on low-resolution grids, and (2) the density of the low-resolution grid to facilitate generalization to other points. It is important to note that numerical integration accuracy does not necessarily imply grid density, especially in the case of low-precision integration formats. We shall experiment on the super-resolution error in Experiment 6.2.

**Choice of Grid**. The choice of grids depends on the specific function and its domain. In the case of a finite interval, a uniform grid is commonly used, and integration can be performed using the trapezoidal rule. Furthermore, selecting a grid that allows for accurate integration effectively captures the function's characteristics at a finite number of points. For instance, in density functional theory (DFT), where the domain is $\mathbb{R}^3$, the function typically involves a multi-center Gaussian mixture. In such cases, designing an integral scheme tailored to multi-center functions is more reasonable. For example, selecting points in the vicinity of each Gaussian center enables better characterization and integration accuracy for the input function. We will conduct the corresponding DFT experiment on molecules in 6.3.

**Extending NOs to Irregular Domains**. The limitation of FNO [19, 23] is its reliance on Fourier bases, which restricts its application to regular domains and limits its usage in real-world complex geometries. Geo-FNO [19] assumes that irregular domains can be mapped to regular ones through a bijection, which is not applicable for arbitrarily irregular domains. Fortunately, under our framework in equation (2), we can overcome this limitation by utilizing random Fourier features (RFFs) and polynomials on irregular domains. By employing Gram-Schmidt orthogonalization, we can obtain an orthogonal basis and perform numerical integration on a discrete grid for the inner product. Moreover, we can select a suitable orthogonal basis based on the domain, such as Gauss-Hermite polynomials for the unbounded whole space $\mathbb{R}$. As a result, we successfully extend NOs to irregular domains, enhancing their applicability in various scenarios. In the first setting of Experiment 6.3, we validate our general NOs on unbounded domains.

**Guiding the Choice of Orthogonal Basis**. The choice of basis in NOs significantly impacts their expressiveness, as highlighted by Theorem 3.1. If the target function can be well represented by a finite combination of basis functions, a small number of modes ($N_{\text{modes}}$) is sufficient. This leads to fewer parameters and increased model efficiency. Conversely, if an infinite series of basis functions is needed to expand the target function, the model tends to generalize poorly. For example, Fourier bases are suitable for periodic functions, wavelets excel in capturing rapid changes and discontinuities, and polynomials (e.g., orthogonal Legendre polynomials) provide a versatile basis for all functions. These bases have distinct capabilities and cannot efficiently represent each other. Additionally, wavelets are effective in handling multi-scale and multi-physics problems. Combining multiple basis sets can yield superior results by leveraging complementary effects. The selection of the basis is guided by the characteristics of the function and the operator in the dataset. For instance, if the function exhibits periodicity in certain subdomains but not in others, a combination of Fourier and polynomial basis functions can effectively model both parts simultaneously, which is verified in Experiment 6.4.

## 5   Related Work

**Orthogonal Basis in NOs**. Various orthogonal bases are adopted for kernel transform in neural operators, e.g., Fourier NO (FNO) [20] and its variant [25, 27, 30] adopt Fourier bases, Geo-FNO [19] utilizes Fourier bases under deformation, [21] uses orthogonal Legendre polynomials, and [28, 8] use multi-wavelets.

**NOs on Arbitrary Domains**. Some operator networks different from NO are grid-free. DeepONet [22] encodes the input function and the grid, respectively, and then combines them by dot product as the operator network output. MIONet [12] extends DeepONet to multiple input functions. Transformer operators [3, 18, 9] directly process the inputs by the attention mechanism [26]. The NO we proposed can also be applied to any domain and incorporates its prior knowledge, so our NO generally outperforms these two approaches.

**Theory of Neural Operators**. DeepONet and FNO are analyzed theoretically in the literature. Theory on DeepONet [16, 7] relies on the discretization over function and the input grid to transform the model into mapping between finite-dimensional vector space. The generalization theory on FNO [14] relies on discretization and proposes Rademacher complexity bounds. [15] proves the universal approximation and errors bound for approximating Darcy type elliptic PDE and the incompressible Navier-Stokes equations. [5, 6, 23] proposes theories for NO's approximation. [4] proves convergence rates for linear operators.

## 6   Experiment

### 6.1   Validation of Theorem 3.1

This section provides empirical evidence demonstrating the superior tightness and quality of our generalization bound presented in Theorem 3.1 compared to related works such as [14]. It is customary in the literature to compare the numerical values of various generalization bounds as a means to showcase their tightness, e.g., [24, 1, 11, 2]. In Figure 1, we provide numerical values of the generalization bounds for FNO models trained on four distinct datasets (1D Burgers, 2D Darcy Flow, 2D+time Navier-Stokes equation, 3D Navier-Stokes equation) as provided by FNO [20]. Following FNO's experimental setup, we normalize the value of our proposed bound to 1 for clarity. Remarkably, our robustness-based bound outperforms existing bounds by 2-3 orders of magnitude, underscoring its superior tightness and reliability.

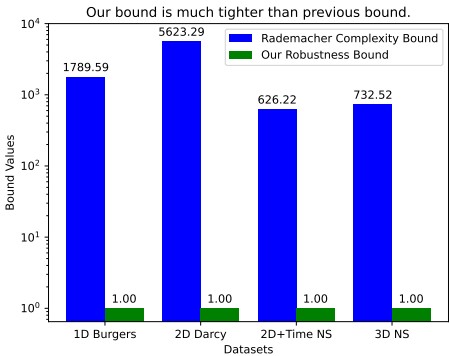

Figure 1: In the four datasets, our generalization bounds (green) are tighter by 2-3 orders of magnitude compared to [14] (blue).

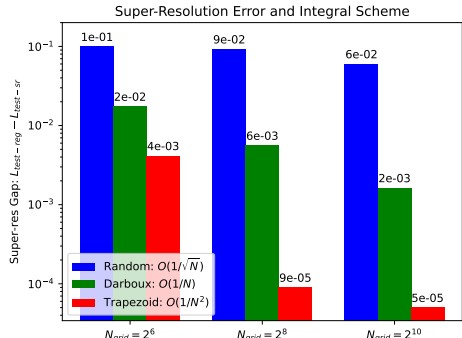

Figure 2: The relationship between super-resolution error, integration format, and the number of integration grid points.

### 6.2   Super-resolution Error

We validate Theorem 3.3 that super-resolution error is affected by both the integration scheme and the grid size. We conduct experiments on the previously mentioned Burgers dataset from FNO

Table 1: Relative $L^2$ error results on DFT datasets with unbounded domains. ANO achieves the best result and can conduct SR.

| | FNO | Geo-FNO | DeepONet | LT | OFormer | NO-Ours |
|---|---|---|---|---|---|---|
| QHO-vanilla | / | 1.22E-2 | 2.63E-3 | 3.04E-3 | 3.31E-3 | **1.50E-3** |
| QHO-superres | / | 1.32E-1 | 1.42E+0 | 7.15E-3 | 7.91E-3 | **2.02E-3** |
| $CO_2$-vanilla | / | 4.97E-1 | 5.82E-1 | 2.46E-1 | 2.41E-1 | **2.22E-1** |
| $CO_2$-superres | / | 5.44E-1 | 1.44E+0 | 2.51E-1 | 2.47E-1 | **2.23E-1** |
| Water-vanilla | / | 2.41E-1 | 3.22E-1 | 3.03E-1 | 2.86E-1 | **1.52E-1** |
| Water-superres | / | 5.07E-1 | 1.75E+0 | 4.52E-1 | 3.07E-1 | **1.58E-1** |
| $CH_4$-vanilla | / | 3.16E-1 | 5.19E-1 | 2.76E-1 | 2.62E-1 | **2.05E-1** |
| $CH_4$-superres | / | 3.95E-1 | 1.82E+0 | 2.81E-1 | 2.79E-1 | **2.07E-1** |

[20], where we use the random scheme (error: $1/\sqrt{N}$, blue), Darboux rule based on a uniform grid (error: $1/N$, green), and trapezoidal rule based on a uniform grid (error: $1/N^2$, red) as integration schemes in FNO [20]. The grid sizes are chosen to be $2^6$ (first column), $2^8$ (second column), $2^{10}$ (third column) points. The super-resolution performance is evaluated by the super-resolution gap $L_{\text{test-reg}} - L_{\text{test-sr}}$ originally defined in Theorem 3.3. Figure 2 illustrates the following findings: (1) increasing the grid size for the same integration scheme improves super-resolution performance, and (2) for the same grid size, integration schemes with higher accuracy yield lower super-resolution error. These results confirm the validity of our theory. It is important to note that the original FNO framework cannot operate on random grids. Our reinterpretation of NOs, utilizing random grids as the integration format and Fourier bases, enabled this capability.

## 6.3 NOs on Unbounded Domain

In this subsection, we select several Density Functional Theory (DFT) Hamiltonian operators to test our NOs with suitable orthogonal bases with other strong baselines.

**Baselines**. (1) FNO [20]: cannot be adopted on an arbitrary domain. (2) Geo-FNO [19]: uses a bijection to map the irregular domain to a regular one and perform FNO. Here the unbounded domains can be mapped to the regular domain by the bijection $\tan^{-1}$, which can conduct super-resolution. (3) DeepONet [22] can operate on arbitrary domains but accepts fixed-length discretized input function, which restricts its super-resolution performance. (4) Linear Transformer (LT) [3] and (5) OFormer [18] are all transformers for operator learning, they can handle arbitrary domains and grids.

In Density Functional Theory (DFT), the Hamiltonian operator plays a crucial role in characterizing the ground state energy through its spectra. In this subsection, we evaluate different NOs' performance in learning Hamiltonian operators defined on unbounded domains across various dimensions.

**QHO**. In the quantum harmonic oscillator (QHO), the Hamiltonian operator, given the wave function $\phi$, is $\left(\hat{H}_{\text{QHO}}\phi\right)(x) = -\frac{1}{2}\nabla^2\phi(x) + \frac{1}{2}x^2\phi(x), x \in \mathbb{R}$. We use random linear combinations of the 1D Hermite polynomial for data generation: $\phi_i(x) = \frac{1}{\pi^{\frac{1}{4}}2^{\frac{i}{2}}\sqrt{i!}}H_i(x)e^{-\frac{x^2}{2}}, x \in \mathbb{R}$, The grid for training is the points in the Gauss-Hermite quadrature with 32 points, and the super-resolution testing grid is with degree 64. Geo-FNO adopts the grid generated by the $\tan^{-1}$ transform of uniform grid over $(-\frac{\pi}{2}, \frac{\pi}{2})$. The orthogonal basis in our NO is Gauss-Hermite polynomials.

**Molecules**. Following D4FT [17], we consider Hamiltonian operators in real-world 3D molecules, which take the wave function $\phi(\boldsymbol{r})$, and the density $\rho(\boldsymbol{r})$ as inputs is given by four different terms (kinetic, external potential, Hartree and exchange-correlation):

$$\left(\hat{H}_{\text{KS-DFT}}(\phi, \rho)\right)(\boldsymbol{r}) = -\frac{1}{2}\nabla^2\phi(\boldsymbol{r}) + v_{\text{ext}}(\boldsymbol{r})\phi(\boldsymbol{r}) + \int_{\mathbb{R}^3}\frac{\rho(\boldsymbol{r}')}{|\boldsymbol{r} - \boldsymbol{r}'|}d\boldsymbol{r}'\phi(\boldsymbol{r}) + v_{\text{xc}}(\boldsymbol{r})\phi(\boldsymbol{r}), \boldsymbol{r} \in \mathbb{R}^3. \quad (6)$$

KS-DFT offers natural basis sets that can be linearly combined to generate the training functions. Additionally, grids with varying resolutions, characterized by levels, are provided based on the multi-center Gaussian nature of the functions in DFT. In our experiments, we select level 1 for training and level 2, which has more points, for testing. In Geo-FNO, the grid is generated using the $\tan^{-1}$ transform of a uniform grid over the domain $(-\frac{\pi}{2}, \frac{\pi}{2})^3$. To ensure a fair comparison, the

Table 2: Relative $L^2$ error results for the advection case.

|  | NO-Sin/Sin | NO-Poly/Poly | NO-Sin/Poly |
|---|---|---|---|
| Advection (1) | **8.34E-3** | 1.96E-2 | 1.01E-2 |
| Advection (2) | 1.00E-2 | 1.76E-2 | **7.66E-3** |

number of grid points in Geo-FNO is similar to the quadrature used in D4FT. For the D4FT setting, we consider three molecules: $CO_2$, water, and $CH_4$.

**Results**. Relative $L^2$ errors for DFT experiments are shown in Table 1. (1) NO-Ours exhibits slightly superior performance on regular testing and significantly outperforms in super-resolution tasks. (2) The uniform grid deformation in Geo-FNO is less efficient compared to quadrature, resulting in NO-Ours surpassing Geo-FNO. (3) As the molecule size increases, atom distribution extends throughout the $\mathbb{R}^3$ domain. Consequently, the input wave function and density function disperse near the atoms rather than concentrating near the unit box. Thus, using a unit box grid in Geo-FNO becomes inefficient. (4) NO-Ours adapts to various grids based on the problem, utilizing efficient quadrature points in D4FT to accurately represent the input wave function and density. As a result, NO-Ours achieves super-resolution with minimal additional error. This highlights the significance of selecting appropriate integral points based on the characteristics of the input functions.

## 6.4 Conbining Multiple Bases

We try the advection equation $u_t + u_x = 0, x \in [0,1], t \in [0,1]$ with/without periodic boundary conditions taken from [23]. Given the initial condition $u_0(x)$, we aim to learn the non-linear operator $\mathcal{G} : u_0(x) \mapsto u(x,t)^2, (x,t) \in [0,1] \times [0,1]$ with the hybrid input functions $u_0(x) = h_1 1_{\{c_1 - \frac{w}{2}, c_1 + \frac{w}{2}\}} + \sqrt{\max(h_2^2 - a^2(x - c_2)^2, 0)}$ where $c_1, c_2, w, h_1, h_2$ are randomly chosen to generate samples. The full problem is named Advection (1), which is periodic in both $t$ and $x$ axis, i.e., $u(x,0) = u(x,1)$ and $u(0,t) = u(1,t)$. To construct a non-periodic problem, we truncate the target function on $(x,t) \in [0,1] \times [0,0.5]$, so that it is periodic on the $x$-axis but not the $t$-axis. We use sinusoidal and/or polynomial bases. Specifically, there are two axes, for NO-Sin/Sin, we use sinusoidal bases on both axes; for NO-Poly/Poly, we use Legendre polynomial bases on both axes; for NO-Sin/Poly, we use sinusoidal on the $x$-axis and polynomial on $t$-axis. More specifically, if we have a set of basis functions $\{\phi_i(x)\}_{i=0}^{\infty}$ on the domain $\Omega_x$ along the $x$-axis, and another set of basis functions $\{\psi_j(t)\}_{j=0}^{\infty}$ on the domain $\Omega_t$ along the $t$-axis, then the set of tensor product basis functions $\{\phi_i(x)\psi_j(t)\}_{i,j=0}^{\infty}$ forms a basis for the two-dimensional space $\Omega_x \times \Omega_t$. We train all models with 1000 epochs.

Table 2 verifies the effectiveness of sinusoidal bases for periodic functions and polynomials for general non-periodic functions. Additionally, the combination of multiple different bases has been shown to be effective, taking into account the properties of the data. This approach deviates from previous works that typically focus on utilizing a single basis.

## 7 Conclusion

This paper proposes a novel perspective for studying NOs. We have provided a comprehensive understanding of NO through a detailed analysis of the infinite sequence space under orthogonal basis projection. Based on this versatile framework, we have proposed a method to adapt NO to arbitrary complex domains. We have also analyzed the generalization bound of NO, demonstrating its superiority over previous works. Furthermore, we have explained the importance of selecting the type and quantity of basis functions in NO, emphasizing the benefits of using multiple bases in a complementary manner based on operator characteristics. Additionally, we have examined the impact of grid points on super-resolution error, highlighting the crucial role of the integration format associated with the grid and the density of the grid itself. All the theoretical analyses have been extensively validated through experiments on multiple data sources, including numerical PDE and DFT. We shed new light on understanding NOs and improving them in practical applications.

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
