## A Experimental Details

### A.1 Density Functional Theory in Experiment 6.3

For D4FT experiments, we choose 4-layer neural operators and train them for 2k epochs. The width of Geo-FNO is 32, while FNO and NO's is 64, for similar running time and parameter number. The number of modes of all models is the number of atomic orbitals to align with NO, i.e., the number of modes in NO is just the number of atomic orbitals in the molecule, which forms the orthogonal basis set. The number of modes of Geo-FNO depends on the grid size for Geo-FNO, where the grid size is small due to high-dimension, which is at most $\lfloor N_{\text{grid}}/2 \rfloor + 1$.

## B Proof of Theory

### B.1 Proof of Proposition 3.1

*Proof.* (Proof of Proposition 3.1) Consider all $c, c' \in l^2$ with $c_i = \langle f, \phi_i \rangle, c_i' = \langle f', \phi_i \rangle$ for the input functions $f, f'$, and also $\Sigma c_i = \langle \sigma(f), \phi_i \rangle, \Sigma c_i' = \langle \sigma(f'), \phi_i \rangle$ where $\sigma$ is the nonlinear activation in the function space while $\Sigma$ is that in the coefficient space,

$$
\left( \sum_{i=0}^{\infty} \| \Sigma c_i - \Sigma c_i' \|_2^2 \right)^{1/2} = \| \sigma(f) - \sigma(f') \|_{L^2} \le L \| f - f' \|_{L^2} = L \left( \sum_{i=0}^{\infty} \| c_i - c_i' \|_2^2 \right)^{1/2} ,
\tag{7}
$$

using Parseval's theorem. $\qquad\square$

### B.2 Proof of Theorem 3.1

We denote $\mathcal{A}$ as the input space and $\mathcal{Z}$ as the input and label pair space.

**Definition B.1.** *[29] For a metric space $(S, \rho)$ and $T \subset S$ we say that $\hat{T} \subset S$ is an $\epsilon$-cover of $T$, if $\forall t \in T$, there $\exists \hat{t} \in \hat{T}$ such that $\rho(t, \hat{t}) \le \epsilon$. The $\epsilon$-covering number of $T$ is defined as:*

$$
\mathcal{N}(\epsilon, T, \rho) = \min\{|\hat{T}| : \hat{T} \text{ is an } \epsilon - cover \text{ of } T\}.
\tag{8}
$$

**Definition B.2.** *[29] A learning algorithm $\mathcal{A}$ is $(K, \epsilon(\cdot))$-robust, for $K \in \mathbb{N}$ and $\epsilon(\cdot) : \mathcal{Z}^n \mapsto \mathbb{R}$, if $\mathcal{Z}$ can be partitioned into $K$ disjoint sets, denoted by $\{\mathcal{C}_k\}_{k=1}^K$, such that the following holds for all $S \in \mathcal{Z}^n$:*

$$
\forall s \in S, \forall z \in \mathcal{Z}, \forall k = 1, \dots, K : \text{ if } s, z \in \mathcal{C}_k, \text{ then } |\ell(\mathcal{A}_S, s) - \ell(\mathcal{A}_S, z)| \le \epsilon(S).
\tag{9}
$$

**Lemma B.1.** *[29] If a learning algorithm $\mathcal{A}$ is $(K, \epsilon(\cdot))$-robust, then for any $\delta > 0$, with probability at least $1 - \delta$ over an iid draw of $n$ examples $S = (z_i)_{i=1}^n$, the following holds:*

$$
\left| \mathbb{E}_z \left[ \ell(\mathcal{A}_S, z) \right] - \frac{1}{n} \sum_{i=1}^n \ell(\mathcal{A}_S, z_i) \right| \le \epsilon(S) + M \sqrt{\frac{2K \ln 2 + 2 \ln(1/\delta)}{n}},
\tag{10}
$$

*where $M$ is defined as follows: for all $h \in \mathcal{H}$ and $z \in \mathcal{Z}$, the loss is upper bounded by $M$ as $\ell(h, z) \le M$.*

**Lemma B.2.** *(Robustness of NO) Consider the NO given by equation (2) with the parameters $\{B_l, W_l\}_{l=0}^L$ and its activation function $\sigma$ is 1-Lipschitz. Then, the mapping is $\left( \mathcal{N}(\gamma/2, \mathcal{A}, \|\cdot\|_{L^2}), \prod_{l=1}^L (\max\{\|B_{l,i} + W_l\|_2, \|W_l\|_2\}) \gamma \right)$-robust for all chosen $\gamma > 0$.*

*Proof.* Recall the layer of NO:

$$
\begin{aligned}
\hat{c}_{l,i} &= \Sigma c_{l,i}, \quad l \ge 1; \\
c_{l+1, \le N_{\text{modes}}} &= (B_{l, \le N_{\text{modes}}} + W_l) \hat{c}_{l, \le N_{\text{modes}}}, \quad c_{l+1, > N_{\text{modes}}} = W_l \hat{c}_{l, > N_{\text{modes}}}; \\
v_i &= c_{L,i};
\end{aligned}
\tag{11}
$$

490 Since, the mapping $\Sigma$ in the first line is 1-Lipschitz,

$$
\begin{aligned}
\|c^{l+1} - d^{l+1}\|_{l^2}^2 &= \sum_{i=0}^{N_{\text{modes}}} \|c_i^{l+1} - d_i^{l+1}\|_2^2 + \sum_{i=N_{\text{modes}}}^{\infty} \|c_i^{l+1} - d_i^{l+1}\|_2^2 \\
&\leq \sum_{i=0}^{N_{\text{modes}}} \|\boldsymbol{B}_l^i + \boldsymbol{W}_l\|^2 \|c_i^l - d_i^l\|_2^2 + \|\boldsymbol{W}_l\|_2^2 \sum_{i=N_{\text{modes}}}^{\infty} \|c_i^l - d_i^l\|_2^2 \\
&\leq \max \left\{ \|\boldsymbol{B}_{l,i} + \boldsymbol{W}_l\|_2, \|\boldsymbol{W}_l\|_2 \right\}^2 \|c^l - d^l\|_{l^2}^2
\end{aligned}
\tag{12}
$$

491 Consequently, the robustness of the $l$th layer depends on $\max \left\{ \|\boldsymbol{B}_{l,i} + \boldsymbol{W}_l\|_2, \|\boldsymbol{W}_l\|_2 \right\}$, while that
492 of the entire model will be $\prod_{l=0}^{L} \max \left\{ \|\boldsymbol{B}_{l,i} + \boldsymbol{W}_l\|_2, \|\boldsymbol{W}_l\|_2 \right\}$. $\qquad\square$

493 *Proof.* (Proof of Theorem 3.1) It immediately follows from Lemmas B.1 and B.2. $\qquad\square$

## B.3 Proof of Theorem 3.3

495 *Proof.* We consider a uniform grid over bounded regular domains. Thus, although the grid points are
496 not randomly chosen, the robustness bound still holds. We consider one layer of NO,

$$
\begin{aligned}
\|u_{l+1}(\boldsymbol{x}) - u_{l+1}(\boldsymbol{x}')\| &= \left\| \int [K(\boldsymbol{B}_l, \boldsymbol{x}, \boldsymbol{y}) - K(\boldsymbol{B}_l, \boldsymbol{x}', \boldsymbol{y})] \hat{u}_l(\boldsymbol{y}) d\boldsymbol{y} + W_l [\hat{u}_l(\boldsymbol{x}) - \hat{u}_l(\boldsymbol{x}')] \right\| \\
&= \left\| \sum_{i=0}^{N_{\text{modes}}} \boldsymbol{B}_{l,i} \langle \hat{u}_l, \phi_i \rangle [\phi_i(\boldsymbol{x}) - \phi_i(\boldsymbol{x}')] + W_l [\hat{u}_l(\boldsymbol{x}) - \hat{u}_l(\boldsymbol{x}')] \right\| \\
&\leq \sum_{i=0}^{N_{\text{modes}}} \|\boldsymbol{B}_{l,i} \langle \hat{u}_l, \phi_i \rangle\| \|\phi_i(\boldsymbol{x}) - \phi_i(\boldsymbol{x}')\| + \|W_l\| \|u_l(\boldsymbol{x}) - u_l(\boldsymbol{x}')\|
\end{aligned}
\tag{13}
$$

497 By induction, we obtain

$$
\begin{aligned}
&\|v(\boldsymbol{x}) - v(\boldsymbol{x}')\| \\
&\leq \sum_{i=0}^{N_{\text{modes}}} \|\boldsymbol{B}_{L-1,i} \langle \hat{u}_{L-1}, \phi_i \rangle\| \|\phi_i(\boldsymbol{x}) - \phi_i(\boldsymbol{x}')\| + \|W_{L-1}\| \|u_{L-1}(\boldsymbol{x}) - u_{L-1}(\boldsymbol{x}')\| \\
&\leq \sum_{i=0}^{N_{\text{modes}}} \sum_{l=0}^{L-1} \left( \prod_{k=l+1}^{L-1} \|W_k\| \right) \|\boldsymbol{B}_{l,i} \langle \hat{u}_l, \phi_i \rangle\| \|\phi_i(\boldsymbol{x}) - \phi_i(\boldsymbol{x}')\| + \prod_{l=0}^{L-1} \|W_l\| \|f(\boldsymbol{x}) - f(\boldsymbol{x}')\|,
\end{aligned}
\tag{14}
$$

498 where $f$ is the input and $v$ is the output. Consequently, the Lipschitz constant of the output function $v$
499 can be bounded by:

$$
\sum_{i=0}^{N_{\text{modes}}} \sum_{l=0}^{L-1} \left( \prod_{k=l+1}^{L-1} \|W_k\| \right) \|\boldsymbol{B}_{l,i} \langle \hat{u}_l, \phi_i \rangle\| \text{Lip}(\phi_i) + \prod_{l=0}^{L-1} \|W_l\| \text{Lip}(f).
\tag{15}
$$

500 $\qquad\square$

## B.4 Proof of Theorem 3.2

502 *Proof.* (Proof of Theorem 3.2) We consider the $l$th layer of the continuous NO,

$$
\begin{aligned}
u_{l+1}(\boldsymbol{x}) &= \int K(\boldsymbol{B}_l, \boldsymbol{x}, \boldsymbol{y}) \hat{u}_l(\boldsymbol{y}) d\boldsymbol{y} + \boldsymbol{W}_l \hat{u}_l(\boldsymbol{x}) \\
&= \sum_{i=0}^{N_{\text{modes}}} \boldsymbol{B}_{l,i} \phi_i(\boldsymbol{x}) \left( \int \hat{u}_l(\boldsymbol{y}) \phi_i(\boldsymbol{y}) d\boldsymbol{y} \right) + \boldsymbol{W}_l \hat{u}_l(\boldsymbol{x}).
\end{aligned}
\tag{16}
$$

Denote the discretized version of the model with $U_l(\boldsymbol{x})$, where

$$
\begin{aligned}
U_{l+1}(\boldsymbol{x}) &= \hat{\int} K(\boldsymbol{B}_l, \boldsymbol{x}, \boldsymbol{y})\hat{U}_l(\boldsymbol{y})d\boldsymbol{y} + \boldsymbol{W}_l\hat{U}_l(\boldsymbol{x}) \\
&= \sum_{i=0}^{N_{\text{modes}}} \boldsymbol{B}_{l,i}\phi_i(\boldsymbol{x})\left(\hat{\int}\hat{U}_l(\boldsymbol{y})\phi_i(\boldsymbol{y})d\boldsymbol{y}\right) + \boldsymbol{W}_l\hat{U}_l(\boldsymbol{x}),
\end{aligned}
\tag{17}
$$

where $\hat{\int}$ denotes the numerical integral. The difference is

$$
\begin{aligned}
u_{l+1}(\boldsymbol{x}) - U_{l+1}(\boldsymbol{x}) &= \sum_{i=0}^{N_{\text{modes}}} \boldsymbol{B}_{l,i}\phi_i(\boldsymbol{x})\left(\left(\int - \hat{\int}\right)\hat{U}_l(\boldsymbol{y})\phi_i(\boldsymbol{y})d\boldsymbol{y}\right) + \\
&\quad \sum_{i=0}^{N_{\text{modes}}} \boldsymbol{B}_{l,i}\phi_i(\boldsymbol{x})\left(\int\left(\hat{u}_l(\boldsymbol{y}) - \hat{U}_l(\boldsymbol{y})\right)\phi_i(\boldsymbol{y})d\boldsymbol{y}\right) + \\
&\quad \boldsymbol{W}_l\left(\hat{u}_l(\boldsymbol{x}) - \hat{U}_l(\boldsymbol{x})\right).
\end{aligned}
\tag{18}
$$

Expand the function $\hat{u}_l(\boldsymbol{x}) - \hat{U}_l(\boldsymbol{x})$ by the orthogonal basis $\phi_i(\boldsymbol{x})$:

$$
\begin{aligned}
u_{l+1}(\boldsymbol{x}) - U_{l+1}(\boldsymbol{x}) &= \sum_{i=0}^{N_{\text{modes}}} \boldsymbol{B}_{l,i}\phi_i(\boldsymbol{x})\left(\left(\int - \hat{\int}\right)\hat{U}_l(\boldsymbol{y})\phi_i(\boldsymbol{y})d\boldsymbol{y}\right) + \\
&\quad \sum_{i=1}^{\infty} \left(\boldsymbol{B}_{l,i} + \boldsymbol{W}_l\right)\phi_i(\boldsymbol{x})\left(\int\left(\hat{u}_l(\boldsymbol{y}) - \hat{U}_l(\boldsymbol{y})\right)\phi_i(\boldsymbol{y})d\boldsymbol{y}\right),
\end{aligned}
\tag{19}
$$

where we denote $\boldsymbol{B}_i^l = 0$ for $i > N_{\text{modes}}$. Therefore,

$$
\|u_{l+1} - U_{l+1}\|_{L^2} \leq \sum_{i=0}^{N_{\text{modes}}} \|\boldsymbol{B}_{l,i}\|_2 e_{\text{grid}}(N_{\text{grid}})e_{\text{func}}\left(\hat{U}_l \cdot \phi_i\right) + \max\left\{\|\boldsymbol{B}_{l,i} + \boldsymbol{W}_l\|_2, \|\boldsymbol{W}_l\|_2\right\}\|u_l - U_l\|_{L^2}.
\tag{20}
$$

For the first layer of the model:

$$
\|u_1 - U_1\|_{L^2} \leq \sum_{i=0}^{N_{\text{modes}}} \|\boldsymbol{B}_{0,i}\|_2 e_{\text{grid}}(N_{\text{grid}})e_{\text{func}}\left(f \cdot \phi_i\right).
\tag{21}
$$

By induction, we can derive the integral error for the entire model:

$$
\|v - V\|_{L^2} \leq \sum_{l=0}^{L} \prod_{k=l}^{L} \max\left\{\|\boldsymbol{B}_{k,i} + \boldsymbol{W}_k\|_2, \|\boldsymbol{W}_k\|_2\right\}\left(\sum_{i=0}^{N_{\text{modes}}} \|\boldsymbol{B}_{l,i}\|_2 e_{\text{grid}}(N_{\text{grid}})e_{\text{func}}\left(\hat{U}_l \cdot \phi_i\right)\right).
\tag{22}
$$

$\square$