# OpenReview forum: "Generalization in Neural Operator: Irregular Domains, Orthogonal Basis, and Super-Resolution"
_NeurIPS.cc/2023/Conference — Submitted to NeurIPS 2023_

### Official Review · Reviewer_59nv · 2023-07-02

**Soundness:** 3 good
**Presentation:** 3 good
**Contribution:** 2 fair
**Rating:** 4
**Confidence:** 3

**Summary:**

This paper studies the generalization error of neural operators that contain kernel operations. Under the basic setting of neural operators (such as FNO), this paper establishes upper bounds of the excess risk of neural operators. How to apply NO to irregular domains, and the error analysis of super resolution are also discussed. The techniques of this paper is rather standard. Some parts of the upper bounds are not clear. I think the contribution of this work is incremental.

**Strengths:**

1. This paper provides an upper bound for the excess risk of neural operators.

2. The upper bound in this paper improves the one in [14]

3. Extension of NOs to irregular domains is discussed.

4. The super resolution error is studied.

**Weaknesses:**

1. The technique used in this paper is pretty standard. The author should emphasis their novelties.

2. Theorems in this paper is not impressive and is unclear. For example, the error bound in Theorem 3.1 depends on the network structure, parameters, and the covering number of the space $B$, which is infinite dimensional. If the norms of network parameters are all larger than 1, the upper bound can be very large. Since the space $B$ is infinite dimensional, the covering number can also be very large, which makes the result less attractive. On the other hand, the magnitude of the training loss is also unclear. I believe the training loss depends on the network's width and depth, which relates closely to the upper bound in this paper. It will make the paper much stronger if these relations are analyzed clearly and a more clear upper bound is derived.

3. For applying NOs to problems with irregular domains, the authors only give an example on unbounded domain. The case for arbitrary irregular domains is only briefly discussed. However, the authors claim the construction of NOs on arbitrary domains as the second contribution. More details on this part should be given.

**Questions:**

1. The upper bound in Theorem 3.1 depends on the covering number of the set $B$, which is an infinite dimensional space. Could the authors discuss how to estimate and bound the covering number?

2. The bounds in this paper are only for excess risks, which also depends on the training loss. Could the author discuss how to bound the training loss?

3. For the super-resolution error, why the upper bound does not depend on N_{grid,test}?

4. In Section 6.1, how the upper bound is computed? The upper bound in this paper requires a tradeoff between \gamma and N_{train}. How this tradeoff is made and how the covering number is computed?

**Limitations:**

Limitations are not disucssed in this paper.

---

> ### Author Rebuttal · Authors · 2023-08-08
>
> Thanks for your valuable comments. Questions are addressed as below.
>
> >Q1: Novelty?
>
> **A1**: See general response.
>
> >Q2: Theorems are not impressive and are unclear...
>
> **A2**: **Network parameter norm larger than 1**. First, it aligns with the effectiveness of our commonly used L2 regularization. Generally, employing regularization tends to lead to smaller norms of model parameters, which in turn correspond to reduced generalization error. Furthermore, our intention is to convey to the readers that by selecting an appropriate basis, the norm of model parameters does not need to be excessively large (meaning the function represented by the model is not overly complex) to achieve a well-fitting of the target function. Thus, simpler models have a better generalization to unknown test data. This is consistent with the performance of various bases in our experiments (Experiment 6.4). Additionally, our derived bound outperforms (Experiment 6.1).
>
> **Covering number of input space $B$**. See A4.
>
> **Training loss**. See our answer to A5.
>
> >Q3: On irregular domains
>
> **A3**: We chose to test our approach on an unbounded domain due to its notably challenging nature which is never tackled before by others. For existing methods, dealing with an entire infinite domain poses considerable difficulties in selecting function value sampling points for model input. But, our methodology, facilitated by the orthogonal basis setup, provides an easily manageable solution for point selection using the quadrature points on unbounded domains, which is in turn strongly correlated with the problem nature and the input functions, validated by our experiment on DFT in section 6.3.
>
> Regarding the conventional irregular domain scenario, due to constraints of paper length, we devoted limited discussion to it. Nevertheless, we underscore the universality of our framework in addressing such cases since an orthogonal basis can be constructed on arbitrary domains (see lines 246-256 on page 6). While the full depth of our treatment may be concise in the current presentation, we assert that our approach is inherently adaptable and applicable to arbitrary irregular domains, representing a significant facet of our secondary contribution.
>
> >Q4: On the covering number of the set $B$
>
> **A4**: Owing to the finite precision at which computers handle input, functions are essentially represented by a series of discrete values. Consequently, the input space is of finite dimensionality, leading to a finite covering number. It is also important to note that our Theorem 3.1 also works for discretized neural operators implemented in practice, by substituting the infinite-dimensional $l^2$ space by the corresponding finite-dimensional Euclidean space, where the input set $B$ has a finite dimension.
>
> When considering continuous function spaces, two primary scenarios arise. Firstly, in cases where high-dimensional data resides on a lower-dimensional manifold, the input set $B$ assumes a lower dimensionality, thus resulting in a finite covering number. Secondly, in situations where the input set $B$ genuinely exists in an infinite-dimensional space and possesses an infinite covering number, our theory accommodates this circumstance by unveiling the impracticality of machine learning within such a context.
>
> To elaborate, when $B$ has an infinite covering number, finite point sampling inadequately represents the distribution of training data. Consequently, any learning procedure grounded in finite training data is insufficient for deducing the inherent characteristics of the operator itself. This scenario embodies the essence of our theoretical framework, serving as an indicator of when operator learning becomes unattainable.
>
> >Q5: On train loss.
>
> **A5**: Fundamentally, this paper provides posterior bounds, and generalization is based on the training loss and model complexity. This setting is widely adopted within the field of statistical learning theory. Concerning the bound for the training loss, it constitutes a distinct consideration from the focus of our investigation. Existing theoretical results demonstrate the universal approximation capabilities of neural operators arxiv.org/abs/2107.07562. Consequently, with appropriate optimization, the training loss can indeed approach minimal values. This assertion of low train loss: arxiv.org/abs/2002.08709.
>
> >Q6: For the super-resolution error, why the upper bound does not depend on $N_{grid,test}$?
>
> **A6**: We assume the true super-resolution error is evaluated using continuous integration, i.e., corresponding to the case where $N_{grid,test}$ tends towards infinity. This represents the true essence of super-resolution error. However, in practical situations, $N_{grid,test}$ is finite, offering an approximation to the continuous integration. We have defined the abovementioned setting in line 83 of page 2. For any given finite $N_{grid,test}$, our theoretical framework can be effectively extended.
>
> >Q7: In Section 6.1, how the upper bound is computed?
>
> **A7**: See **Q4** for bounding the covering number of $B$.
>
> For $\gamma$, we choose it as 0.1, and for the covering number of a discretized finite-dimensional function space, it is the same as computing the covering number of a finite vector space containing all the data points.
>
> Suppose that the input space is normalized to $[0,1]^d$ ($d$ is the input dimension), which is a common practice in deep learning for training stability, we simply need to place $1/\gamma$ different grid points at intervals of $\gamma$ along each dimension of the input space. This set of grid points forms a $\gamma$-cover for the input space, and the cardinality of this set can be calculated.
>
> Furthermore, the techniques in https://arxiv.org/abs/2206.13497 can be adopted to further reduce our bound's numerical value.
>
> Meanwhile, the term $B_{l,i},W_l$ are the trained model parameters. $M$ is the upper bound is the loss function. $N_{train}$ is the number of training data.

---

> > ### Comment · Reviewer_59nv · 2023-08-18
> >
> > I thank the authors for the detailed response. I suggest the authors to add the computation of covering number to the main paper. I still have some question.
> >
> > For Network parameter norm, Experiment 6.4 only provides the experimental error. We still have no idea how large the parameter norm is. I understand that the authors want to show simpler models have better performance. But it is unclear whether small parameter norm will give a small training loss. The two reference provided by the authors do not mention the scale of parameters. But in many existing work that discuss parameter scale, such as https://arxiv.org/abs/1610.01145, the parameters are very large.
> > Consider an extreme case, if we require all parameters are very close to 0, and when N_{train} is large enough, your error bound will be almost 0. However, under such a setting, I think the training loss will be very large. Even though we have a very small upper bound, but it does not make any sense. Perhaps I am wrong. Could the author provide some insights or evidence (either theoretical or experiment) on this point?

---

> > > ### Author Response · Authors · 2023-08-19
> > > **Response to the reviewer**
> > >
> > > We would like to thank the reviewer for the helpful discussion. Your further question is addressed below. Let us know if you need more explanation.
> > >
> > > >Q8: I suggest the authors add the computation of the covering number to the main paper
> > >
> > > **A8**: Sure! Thanks for pointing this out. Actually, we refer the reviewer to lines 172-177 in the main paper, where we have already explained it during the submission. We will add more details for sure in the revision.
> > >
> > > >Q9: Consider an extreme case, if we require all parameters to be very close to 0, and when $N_{train}$ is large enough, your error bound will be almost 0. However, in such a setting, I think the training loss will be very large. Even though we have a very small upper bound, it does not make any sense.
> > >
> > > **A9**: The phenomenon pointed out by the reviewers is actually a trade-off between model complexity and lower training error. In practical model training, we often encounter a situation where shorter training times lead to higher training errors, resulting in less optimized models that are comparatively simpler and have smaller parameter matrix norms. On the other hand, longer optimization times tend to reduce training errors to near zero, but the resulting models can be more complex and may even lead to overfitting, corresponding to larger parameter matrix norms. Thus, this situation represents a trade-off.
> > >
> > > In practice, this trade-off is often managed through validation data. We typically set aside around 10\% of the data as validation data. During training, we select the epoch with the lowest validation data error for testing. In theoretical terms, our established bounds precisely describe this trade-off. Specifically, the smaller the training loss, the higher the risk of overfitting, and conversely, a larger training loss may lead to underfitting. Therefore, we need to strike a balance using the validation loss, aiming to identify a middle point where the model performs optimally.
> > >
> > > >Q10: But it is unclear whether a small parameter norm will give a small training loss. The two references provided by the authors do not mention the scale of parameters. But in many existing works that discuss parameter scale, such as arxiv 1610.01145, the parameters are very large.
> > >
> > > **A10**: We greatly appreciate the reviewers for highlighting this issue, as it delves into the enigma of generalization in deep learning. Deep learning is often able to learn intricate models that exhibit the capability to generalize to unseen data, and this paradox has been a subject of significant interest. Dr. Albert Einstein has a famous quote: “In theory, theory and practice are the same. In practice, they are not.” The main objective of theoretical work, in this context, is to facilitate a comparison of the generalization capabilities of two models under identical training conditions, providing a preliminary understanding before conducting tests on unknown datasets.
> > >
> > > For instance, in the models presented in Experiment 6.4, the parameter matrix norms are now supplemented with values as below. This information from the theoretical perspective enables us to discern which model is likely to perform better in terms of generalization, thereby guiding the practical decision-making process on the model choice. Every theory has its applicable scope, and our aim is to equip users of the neural operator framework with insights into which model could exhibit superior performance on test data under the same training environment.
> > >
> > > This objective extends to our other theorems as well, such as those concerning discretization and super-resolution errors. Through theoretical analyses, we delineate three influential factors: model complexity, the numerical format of integration, and grid density. These factors have demonstrated remarkable alignment with real-world scenarios. We are confident that our theoretical framework offers superior guidance for comprehending the generalization, super-resolution, and discretization errors intrinsic to neural operators.
> > >
> > > In conclusion, the aim is to endow practitioners with intuitive insights and a mechanism for comparing different models under similar conditions.
> > >
> > > **This related work validates the relation between bound and empirical performance too, serving as a justification: arXiv 2109.09444.**
> > >
> > > Here are the additional results:
> > >
> > > ===== Model performance in Table 2 of the original paper =====
> > >
> > >          NO-Sin/Sin || NO-Poly/Poly || NO-Sin/Poly
> > >
> > > Advection (1) ||   8.34E-3  ||     1.96E-2      ||      1.01E-2
> > >
> > > Advection (2) ||   1.00E-2  ||     1.76E-2      ||      7.66E-3
> > >
> > >
> > > ===== Computed bound for the models =====
> > >
> > >
> > >            NO-Sin/Sin || NO-Poly/Poly || NO-Sin/Poly
> > >
> > > Advection (1) ||   100%      ||     259%         ||      173%
> > >
> > > Advection (2) ||   100%      ||     195%         ||      82%
> > >
> > > Here we normalize the bound of the first model NO-Sin/Sin as 100% for clear comparison.
> > >
> > > As you can see, the theoretical bound and the empirical results are consistent.

---

> > > ### Author Response · Authors · 2023-08-21
> > > **Looking forward to more discussion**
> > >
> > > Dear Reviewer 59nv,
> > >
> > > We want to express our thanks once more for your comprehensive and perceptive input. We have addressed your additional considerations. **As we approach the end of this phase of response, we are looking forward to your further response.**
> > >
> > > Specifically, **we summarize our response for you, for the full version kindly refer to our previous response.**
> > >
> > > >Q9: Consider an extreme case, if we require all parameters to be very close to 0, and when $N_{train}$ is large enough, your error bound will be almost 0. However, in such a setting, I think the training loss will be very large. Even though we have a very small upper bound, it does not make any sense.
> > >
> > > **A9**: This scenario reflects a trade-off between model complexity and lower training error. Shorter training can lead to higher training errors but simpler models with smaller parameter norms, while longer training reduces the training errors but can result in complex models and overfitting. Validation data helps strike a balance. Our bound can reflect the tradeoff and is thus informative.
> > >
> > > >Q10: But it is unclear whether a small parameter norm will give a small training loss. The two references provided by the authors do not mention the scale of parameters. But in many existing works that discuss parameter scale, such as arxiv 1610.01145, the parameters are very large.
> > >
> > > **A10**: Theoretical insights aid in comparing generalization capabilities under identical conditions, guiding model selection. Models in Experiment 6.4 illustrate the relationship between parameter norms and generalization: Additional numerical results also show that our theoretical bounds align with empirical results.
> > >
> > > Consider a practical scenario involving the application of a neural operator. In this context, the nature of the testing dataset remains unknown, and prior to its deployment into production, our derived bounds can be employed to facilitate the process of model selection.
> > >
> > > **Here is an important reference that shows that the theoretical bound based on the parameter matrix norm basically aligns with the empirical performance in most cases:**
> > >
> > > When Do Extended Physics-Informed Neural Networks (XPINNs) Improve Generalization? SIAM Journal on Scientific Computing (SISC).
> > > arXiv 2109.09444
> > >
> > > The authors of that paper validate the bounds of physics-informed neural networks (PINNs) for solving partial differential equations (PDEs), which is closely related to neural operators solving PDEs too.
> > >
> > > Another justification for the parameter matrix norm-based bounds in our paper is that the bounds can help inspire new regularization methods to prevent overfitting for deep neural operators. L2 and L1 regularizations are proven to be successful in deep learning, which is closely related to the theory in the related work (see arXiv 1712.06541). One related work designed novel regularization based on the parameter matrix norm-based they derived (see arXiv 2205.11359).
> > >
> > > In sum, our bound provides a means to compare the different models to anticipate their test performance before real-world deployment and inspires novel regularization for neural operators, and the bounds are consistent with the practice in most cases.
> > >
> > > Warm regards,
> > >
> > > The authors

---

> > > ### Author Response · Authors · 2023-08-21
> > > **Last message before rebuttal ends**
> > >
> > > Dear Reviewer 59nv,
> > >
> > > Hope you can refer to our rebuttal at the decision stage. We are grateful for your helpful discussion.
> > >
> > > Warm regards,
> > >
> > > The authors

---

### Official Review · Reviewer_XCTm · 2023-07-03

**Soundness:** 3 good
**Presentation:** 3 good
**Contribution:** 3 good
**Rating:** 7
**Confidence:** 2

**Summary:**

A theoretical analysis of neural operators (NOs) is presented, which provides further insight into the construction and performance of NOs.  The theoretical insights are validated by numerical experiments.

**Strengths:**

A thorough theoretical analysis of NOs is presented, with significant improvements in the tightness of generalization bounds compared to prior work.  The implications of the theory and insights that may be drawn are discussed.  While some of these may seem obvious, it is important to base such insight on robust underlying theory, as presented.  The insights gained from the theory are validated by numerical experiments.

**Weaknesses:**

The summary of the overall model is not the most clear (Equation 1), which suggests the non-linear activation is applied first, rather than following the kernel and linear transforms.

Typo: "project" -> "projection"

Typo: sometimes $\mathcal{L}$ used to represent loss, sometimes $L$.

Typo: "Conbining" -> "Combining"

**Questions:**

While Figure 1 shows the generalization bounds are much tighter than alternative work, how can one have confidence the bounds are valid?

**Limitations:**

No special societal concerns.

---

> ### Author Rebuttal · Authors · 2023-08-08
>
> We would like to thank the reviewer for the valuable comments. Your questions are addressed as follows.
>
> >Q1: The summary of the overall model is not the most clear (Equation 1), which suggests the non-linear activation is applied first, rather than following the kernel and linear transforms.
>
> **A1**: In equation (1), our intention is to convey that for the 0-th layer, which corresponds to the input layer, activation is not required for the input to undergo the subsequent kernel transformation and linear transformation. However, for layers beyond the 1st, we first apply activation to the input from the preceding step, followed by the kernel transformation and linear transformation. Hence, we emphasize that activation is needed for $l \geq 1$, and no activation is performed for
> $l=0$. This distinction arises fundamentally from the fact that activation is unnecessary for the 0-th layer, while subsequent layers necessitate activation. To enhance clarity regarding the overall structure of the model, we may consider incorporating a diagram in the revision.
>
> >Q2: On the typos.
>
> **A2**: Thanks for pointing out our typo, we will carefully check all the content in the revision to complete high-quality writing
>
> >Q3: While Figure 1 shows the generalization bounds are much tighter than alternative work, how can one have confidence the bounds are valid?
>
> **A3**: Our generalization bound can be explicitly computed based on the model parameters, the quantity of training data, and the characteristics of the training data. Therefore, the results presented here are a manifestation of this explicit computation. Importantly, our bound is remarkably tight owing to the substantial enhancement we have achieved in reducing the dependency of the bound on model parameters.
>
> In particular, in terms of parameter matrix norm contributed by each layer, please see lines 224-231 in the main paper, which demonstrates that our bound is much tighter than previous work.

---

> > ### Comment · Reviewer_XCTm · 2023-08-16
> > **Response to Authors**
> >
> > Many thanks to the Authors for their response.  I appreciate the generalization bound can be directly computed and compared to other works.  I was more interest in whether there has been any validation of the bound, e.g. mathematical consistency checks, derived by an alternative approach, or numerical validation.  I am not suggesting there is any error but wanted to understand to what extend the results are validated.

---

> > > ### Author Response · Authors · 2023-08-19
> > > **Response to the reviewer**
> > >
> > > We would like to thank the reviewer for the helpful discussion. Your further question is addressed below. Let us know if you need more explanation.
> > >
> > > >Q4: I was more interested in whether there has been any validation of the bound, e.g. mathematical consistency checks, derived by an alternative approach, or numerical validation.
> > >
> > > **A4**: The bound in Theorem 3.1 is mathematically rigorous. Our bound constrains the test error on the left side of the inequality, and it encompasses three components: the first is the training error, the second is the model complexity, and the third is a probabilistic term. A model with smaller training errors and lower complexity is more likely to generalize well. A low model complexity results in a smaller numerical value for the complexity term on the right side, thus enhancing its ability to generalize to unseen test data. The third term is a probabilistic factor that includes the probability of the bound holding true, denoted as $1-\delta$. Typically, we select $\delta=0.1$ to ensure a 90\% probability of the bound holding. The stochastic nature of the bound arises from randomly selecting training points from an unknown data distribution.
> > >
> > > The bounds presented in Theorem 3.2 and Theorem 3.3 are also rigorously established in mathematical terms. They pertain to the discretization error and super-resolution error of the neural operator. Consequently, once the model and data are provided, we can employ theoretical analysis to indicate the specific factors upon which these errors depend. This can serve as inspiration for devising models that exhibit smaller errors. Specifically, both of these bounds primarily rely on the accuracy of the numerical integration scheme employed within the neural operator, the density of grid points, and the complexity of the functions represented by the model which in turn depends on the model parameter matrices. If a model represents functions that are more intricate or steep, the accuracy of numerical integration might decrease, leading to an increase in discretization error.

---

### Official Review · Reviewer_PCWZ · 2023-07-06

**Soundness:** 3 good
**Presentation:** 3 good
**Contribution:** 2 fair
**Rating:** 5
**Confidence:** 3

**Summary:**

The image super-resolution (SR) is a recurrently used task, nowadays, since the SR images can improve the accuracy of downstream tasks like object detection. Many proposals rely on heavy Deep Learning models or lightweight models based on efficiently designed architectures. This work studies neural operators based on examining the orthogonal base. The operations proposed in the manuscript, according to their theorems and demonstrations with the appropriate orthogonal bases and the grip points, reduce the time of convergence and make the network adapt faster to the  irregular domains.

**Strengths:**

* The proposal section in the manuscript  is easy to read and follow.
* Theorem 3.2 and 3.3 are properly defined. For theorem 3.3, this will assist in re-planning new proposals for improving image SR accuracy.

**Weaknesses:**

Despite a short evaluation performed that shows a better performance of the proposal, I feel it is not sufficient to validate the generalization ability in the super-resolution task. To this end is necessary to validate with the standard evaluation metrics used in the SR domain. Personally, I feel that more explanation and motivation is needed for equation 3.

Minor error:
The machine learning articles, mostly, are organized Introduction, related works, the proposal, material, and experiments. It should be convenient to use this structure in the manuscript.

**Questions:**

No questions, but receive the weaknesses

**Limitations:**

There are no limitations addressed in the manuscript

---

> ### Author Rebuttal · Authors · 2023-08-08
>
> Thanks for your valuable comments. Your questions are addressed as follows.
>
> >Q1: Despite a short evaluation performed that shows a better performance of the proposal, I feel it is not sufficient to validate the generalization ability in the super-resolution task. To this end is necessary to validate with the standard evaluation metrics used in the SR domain. Personally, I feel that more explanation and motivation are needed for equation 3.
>
> **A1**: In assessing super-resolution error, we indeed adhere to the standard procedure, specifically encapsulated by the definition on the left side of equation (5) in Theorem 3.3.
>
> We would like to stress that this paper focuses on the theory of neural operators (NOs) which learns mappings between functions, but not image classifiers. We would also like to emphasize that we propose a novel and more accurate definition of superresolution in NOs based on orthogonal bases. Our research is conducive to the NO literature that fails to delve into NOs' superresolution before. So, we compare with the standard definition of superresolution in NOs, see Definition 4 and Theorem 8 in related work https://arxiv.org/pdf/2108.08481.pdf (JMLR paper).
>
> On Definition 4 and Theorem 8 in the JMLR paper: These definitions all pertain to the characterization of the super-resolution error. Notably, our definition of super-resolution error diverges significantly from that presented in this paper. Their focus is primarily geared towards the extrapolation of model super-resolution under extreme conditions, specifically when the grid points of integration and the vector size of model inputs both tend towards infinity. This arises from the requirement in super-resolution for broader pointwise evaluations across the grid, leading to an escalation in the dimensions of the input vector. Consequently, their approach hinges on a limit-based perspective.
>
> Actually, our framework maintains a greater degree of flexibility than that of the JMLR paper. We avoid the utilization of such limiting considerations to define the model. Our approach affords us the capability to investigate various influencing factors, such as the impact of model numerical integration precision and the norm of model parameters on the super-resolution error. In their limit-based framework, regardless of the choice of integration scheme, input functions, or model parameters, the super-resolution error converges to zero as the grid size tends to infinity. In other words, they establish the convergence to this limit, but unlike our approach, they do not provide explicit error analyses or quantify the rate of convergence to zero under different settings within their framework.
>
> This super-resolution error quantifies the disparity between test error on a more refined grid $L_{\text{test-sr}}(\theta_S)$ and test error on a sparser grid $L_{\text{test-reg}}(\theta_S)$. The concept of "super-resolution" materializes as a consequence of employing dissimilar grids during testing, thus characterizing the notion of "super-resolution." Our approach aligns with this established definition, rooted in the traditional practices of the field.
>
> Regarding equation (3), it encapsulates our proposed notion of generalization error. This metric serves as an indicator of the model's ability to generalize its learned behaviors beyond the training data, encapsulating the essence of our theoretical framework.
>
> Revisiting equation (3) in the main paper, $L_{\text{test-reg}}(\theta_S)$ is the regular test error without considering super-resolution. $L_{\text{train}}(\theta_S)$ is the train loss. $B_{l,i}$ and $W_l$ are the model parameters in $\theta_S$ where the model structure is defined in equations (1) and (2) in the paper. The bound holds for all $\gamma > 0$ and choose $\gamma = 0.1$ in our numerical experiment. $K$ (see line 166 of the main paper for detai;s) is the $\gamma/2$-covering number of the input space $\mathcal{B}$ under the l2 norm, which can be computed analytically after given the training data set and the value of $\gamma$. $M$ is the upper bound of the loss function which can be set to the maximal train loss value during multiple training instances, and empirically, after proper choice of the learning rate and model initialization, the loss throughout the training is always bounded. $N_{train}$ is the number of training data. $\delta$ defines the probability of our bound holds, and we choose $\delta=0.1$ in the experiment so that our bound has a probability of 0.9 to hold given the random draw of training data.
>
> >Q2: Minor error: The machine learning articles, mostly, are organized into Introduction, related works, the proposal, material, and experiments. It should be convenient to use this structure in the manuscript.
>
> **A2**: Thank you for your suggestion. However, our manuscript diverges from the conventional structure of other papers in the field. We have intentionally chosen to present our content in a distinct manner, aiming to underscore the significance of our theory and its practical applications. As such, we have dedicated a single section to comprehensively outline the practicality of our theory, covering aspects such as generalization, super-resolution error, discretization error, and irregular domains. Each of these aspects is seamlessly connected to specific experiments, demonstrating the thorough validation of each of our claims. This meticulous approach serves to establish the comprehensiveness and robustness of our paper, even though it deviates from the standard organization you mentioned.

---

> > ### Comment · Reviewer_PCWZ · 2023-08-21
> >
> > Dear Authors:
> >  thank you for addressing my review.  After revising other rebuttals, which make clearance, I can say, It is a good contribution in the Neural Operator domain.
> >
> > I have just changed my rating.

---

### Official Review · Reviewer_8nxh · 2023-07-07

**Soundness:** 3 good
**Presentation:** 3 good
**Contribution:** 3 good
**Rating:** 5
**Confidence:** 2

**Summary:**

In this paper, the authors propose to analyze neural operators from the orthogonal bases in the kernel operators, which helps to guide designing kernel operators and choosing grid points, analyzing generalization and super-resolution capabilities, and adapting neural operators to irregular domains.


**Strengths:**

This is an interesting paper providing not specific models but new analysis perspectives and design principles on Neural Operators.


**Weaknesses:**

- I feel there are still gaps between the overall claims, the theoretical results in Section 3, and the implication and experiments in Sections 4&5. For example, the "NOs on Unbounded Domain" section is not what I would expect for "adapting neural operators to irregular domains"; the "Combining Multiple Bases" section seems more like analyzing kernel operators, instead of providing insight and principles in "designing kernel operators".
- The writing is not always good. For example, in the abstract, similar contents reappear three times.


**Questions:**

None.


**Limitations:**

This work is a bit beyond my capability. I think it is an interesting and important work, but I don't feel like it has reached its perfect state.

---

> ### Author Rebuttal · Authors · 2023-08-08
>
> Thanks for your valuable comments. Your questions are addressed as follows.
>
> >Q1: I feel there are still gaps between the overall claims, the theoretical results in Section 3, and the implication and experiments in Sections 4\&5. For example, the "NOs on Unbounded Domain" section is not what I would expect for "adapting neural operators to irregular domains"; the "Combining Multiple Bases" section seems more like analyzing kernel operators, instead of providing insight and principles in "designing kernel operators".
>
> **A1**: On irregular domains: We chose to test our approach on an unbounded domain due to its notably challenging nature which is never tackled before by others. For existing methods, dealing with an entire infinite domain poses considerable difficulties in selecting function value sampling points for model input. But, our methodology, facilitated by the orthogonal basis setup, provides an easily manageable solution for point selection using the quadrature points on unbounded domains, which is in turn strongly correlated with the problem nature and the input functions, validated by our experiment on DFT in section 6.3.
>
> Regarding the conventional irregular domain scenario, due to constraints of paper length, we devoted limited discussion to it. Nevertheless, we underscore the universality of our framework in addressing such cases since an orthogonal basis can be constructed on arbitrary domains (see lines 246-256 on page 6). While the full depth of our treatment may be concise in the current presentation, we assert that our approach is inherently adaptable and applicable to arbitrary irregular domains, representing a significant facet of our secondary contribution.
>
>
> In the context of defining an orthogonal basis on any irregular domain for constructing the neural operator, the methodology is as follows. Firstly, we can establish a set of linearly independent functions on the arbitrary domain, such as varying-degree polynomials and trigonometric polynomials with different frequencies – these are commonly employed bases widely used in interpolation and similar applications. However, these functions are not orthogonal. The subsequent step involves outlining the process of orthogonalization. For this purpose, a numerical integration scheme is introduced on this domain. A common numerical integration approach, for instance, is based on the Monte Carlo method, wherein a sufficient number of points are randomly sampled within the domain, and the function values at these points are employed for estimating the integral. Alternatively, a more accurate quadrature method can be employed, involving partitioning the irregular domain into a series of triangular grids, akin to finite element methods. Subsequently, a quadrature approach is defined within each triangular grid, and the summation of all quadrature computations over these grids constitutes the overall quadrature integration scheme. Leveraging this numerical integration, we can employ the Gram-Schmidt orthogonalization process to orthogonalize any set of linearly independent functions.
>
> On combining multiple bases: In fact, the process of combining multiple bases not only involves analyzing kernel operators but also encompasses the design of novel kernel operators. Specifically, as demonstrated in Experiment 6.4, the features of the target function differ between the x and t axes; the former is periodic while the latter is non-periodic. To accommodate these distinct function characteristics, we utilize polynomial bases on the non-periodic axis and trigonometric bases on the periodic axis. This strategic choice enables the creation of a more effective neural operator. Another illustration of our theoretical framework's design philosophy can be found in the DFT experiment within Section 6.3. Here, we tailor our approach to the unique characteristics of the DFT target function, employing a multi-center Gaussian-type basis that closely aligns with the function's nature, yielding superior results.
>
> In essence, our proposed method serves a dual purpose of both analyzing and designing kernel operators, as exemplified by these instances. By strategically combining various bases, we manifest our approach's capacity for sophisticated analysis and intentional design, thereby substantiating its broader utility and significance.
>
> >Q2: The writing is not always good. For example, in the abstract, similar contents reappear three times.
>
> **A2**: Thank you for your suggestion. We will certainly address this concern and work on refining the writing in our revision. Your feedback is greatly appreciated as we strive to enhance the quality of the manuscript.

---

> > ### Comment · Reviewer_8nxh · 2023-08-21
> >
> > Thank you very much for your detailed responses. The comments and discussions have been very helpful. I decide to remain my original positive rating.

---

### Official Review · Reviewer_2NvK · 2023-07-09

**Soundness:** 4 excellent
**Presentation:** 4 excellent
**Contribution:** 4 excellent
**Rating:** 4
**Confidence:** 4

**Summary:**

The authors consider a new perspective to neural operators, by examining the role of orthogonal bases. The kernel operators are constructed such that their eigenfunctions are predefined orthogonal bases, with the eingenvalues as trainable parameters. That is, a neural operator can be seen as a mapping from input coefficients to output coefficients of the orthogonal basis functions. The authors show several theoretical results using this new perspective, backed by empirical results:
- Improved generalization bounds.
- Super-resolution error bounds.
-  Irregular domains - they show that neural operators can be extended to irregular domains using random Fourier features and polynomials on irregular domains.
- Choosing other orthogonal bases. The authors show that Fourier bases can be combined with wavelet or polynomial bases.

**Strengths:**

This paper can be impactful both because of the four concrete results they show (generalization bounds, super resolution bounds, irregular domains, and choice of orthogonal bases), but also because of the novel perspective of neural operators. In particular, the novel eigenvalue / orthogonal basis-based perspective of neural operators can be a useful view for studying neural operators in the future, both theoretically and empirically.

While several other works have studied neural operators for irregular domains / topologies, the other results by the authors are much more novel. Not much prior research has studied generalization bounds, super resolution bounds, or combining bases before this work.

**Weaknesses:**

- Some of the results are fairly obvious or have only limited empirical value. For example, the super-resolution theorem/experiments, although it is novel to study super resolution, the main punchline is that super-resolution depends on the accuracy of the integration method, and the density of the low-resolution grid (i.e., the results in Figure 2), which is not so surprising.
- Section 4 shows implications of the theory, motivating the experiments in the next section. However, some of these experiments have only tangential ties to the theory. For example, “Guiding the choice of Orthogonal Basis” is justified because Theorem 3.1 impacts their expressiveness. I would argue that we would decide to research the choice of orthogonal basis even if we didn’t see Theorem 3.1 first.
- It would be nice if there was a bit more intuition for each of the proofs, in the main text.
- Did not discuss limitations.
- The authors could have released the code (anonymously during submission).

**Questions:**

- It would be good to discuss the relation of this work to other works on the theory of neural operators (beyond the discussion in Section 5). For example, how does your work relate to the JMLR paper https://arxiv.org/pdf/2108.08481.pdf, e.g. your super-resolution results compared to discretization invariance in Definition 4 and Theorem 8, and your perspective of neural operator compared to the set of neural operators used in their theory, defined in Section 9.1?
- Wavelet bases are not necessarily discretization invariant. The definition used in the above JMLR paper says that we fix a finite set of weights for the architecture, and then we can take any discretization of the domain as input. The FNO accomplishes this by truncating to a fixed number of frequency modes $x$, which stays the same at higher resolutions. But it is not clear how Wavelet basis can satisfy this.
- What are the limitations of your work (see below)?
- Can you add a bit more intuition to all the theoretical results?

**Limitations:**

The authors did not discuss the limitations of their work, and they answered “no” without explanation in the OpenReview paper checklist to that question. The NeurIPS call for papers states that authors can answer no to a checklist question, provided they give a good explanation. But I cannot think of any explanation why an author would be justified in not describing the limitations of their work; I think it is strictly better to do so.

---

> ### Author Rebuttal · Authors · 2023-08-08
>
> Thanks for your valuable comments. Your questions are addressed below.
>
> >Q1: Some of the results are fairly obvious.
>
> **A1**: The implications of our theory for practical applications, particularly in **grid type selection but not grid size**, carry profound significance.
>
> Conventionally, generated data often adhere to uniform grids. However, our theory states that super-resolution error is intricately linked to the chosen integration scheme. Commonly used uniform grid integration yields an error proportional to $(1/N_{grid}^2)$ where $N_{grid}$ is the grid size. Alternatively, employing high-precision quadrature significantly diminishes this error at an exponential rate with increasing grid size. This revelation has substantial implications, as it implies that training an equally accurate neural operator requires fewer grid points on a quadrature grid, thus aiding in reducing computational costs during training.
>
> >Q2: Experiments have only tangential ties to the theory.
>
> **A2**: The theory we propose serves three main purposes. Firstly, it provides theoretical underpinnings that offer assurances for observed phenomena. Secondly, it offers practical recommendations to improve neural operator (NO) training. These recommendations include guidance on grid selection, orthogonal basis selection, and modeling NO on irregular domains. The third is the innovative nature of the theoretical tools themselves.
>
> Despite your concern, theory and practice are related. For instance, the investigation into guiding the choice of an orthogonal basis is justified by Theorem 3.1 on their expressiveness.
>
> However, what makes our approach innovative is that we directly handle functions in infinite-dimensional spaces, while most previous studies discretize the infinite-dimensional input onto finite grids before applying traditional finite-dimensional theories. Our theory is versatile and applies even to discretized models. This innovation opens doors for future research to build upon our framework, advancing the field with more refined and comprehensive theoretical insights.
>
> >Q3: IIntuition for each of the proofs.
>
> **A3**: See the general response.
>
> >Q4: Limitations and codes.
>
> **A4**: We will discuss this in the revision and publish codes.
>
> >Q5: Comparison with JMLR paper.
>
> **A5**: On Definition 4 and Theorem 8 in the JMLR paper: These definitions all pertain to the characterization of the super-resolution error. Notably, our definition of super-resolution error diverges significantly from that presented in this paper. Their focus is primarily geared towards the extrapolation of model super-resolution under extreme conditions, specifically when the grid points of integration and the vector size of model inputs both tend towards infinity. This arises from the requirement in super-resolution for broader pointwise evaluations across the grid, leading to an escalation in the dimensions of the input vector. Consequently, their approach hinges on a limit-based perspective.
>
> In contrast, our framework maintains a greater degree of flexibility. We avoid the utilization of such limiting considerations to define the model. Our approach affords us the capability to investigate various influencing factors, such as the impact of model numerical integration precision and the norm of model parameters on the super-resolution error. In their limit-based framework, regardless of the choice of integration scheme, input functions, or model parameters, the super-resolution error converges to zero as the grid size tends to infinity. In other words, they establish the convergence to this limit, but unlike our approach, they do not provide explicit error analyses or quantify the rate of convergence to zero under different settings within their framework.
>
> On Section 9.1 in the JMLR paper: A significant departure between our formulation and theirs lies in how we delve into understanding the neural operator within the coefficient space. Notably, in their paper, the neural operator continues to be treated as a mapping between infinite-dimensional functions, thereby rendering many theoretical analyses inapplicable to such infinite-dimensional functions. In contrast, we ingeniously employ an orthogonal basis to transform the input functions into an equally infinite-dimensional vector space, albeit with real-valued elements. This strategic move enables the adoption of numerous tools from learning theory since the inputs are now an infinite-dimensional real number vector so that we can nontrivially extend traditional results on finite-dimensional real number vector learning. Consequently, this approach facilitates the derivation of more intricate and insightful theories, which underscores the substantial contribution we make in comparison to their work.
>
> >Q6: Wavelet bases are not necessarily discretization invariant.
>
> **A6**: The discussion on wavelets is refs 8 and 28. In essence, the neural operator corresponding to wavelet or any other basis functions entails computing the projection coefficients of the input function onto this basis through numerical integration. These coefficients are then mapped by the parameters of the neural network. Since the coefficients are derived from numerical integration, the concept of super-resolution involves employing different numerical formats for integration. In this context, the coarse grid entails employing coarse discretization points for numerical integration, while super-resolution necessitates more precise and finer discretization points. So, wavelet basis has no difference with other orthogonal bases, the same theoretical framework can still be applied. Therefore, with a well-defined basis and a numerical integration scheme, the concept of super-resolution can indeed be applied.
>
> >Q7: Intuition to all the theoretical results?
>
> **A7**: See the general response.

---

> > ### Comment · Reviewer_2NvK · 2023-08-17
> >
> > Thank you very much for preparing the rebuttal for my review and the other reviews. I especially appreciate the longer discussion on related work, and the discussion on the intuition and contribution of your theoretical results. I encourage you to add these to the paper. Overall I have a better opinion of your work. However, I have a few remaining questions.
> > - Limitations: a couple reviewers mentioned that there was not a discussion of the limitations, even though the NeurIPS checklist specifically brings up that authors should discuss limitations. You said that you would include limitations in the revised manuscript. Please be more specific: what will you say about the limitations of this work?
> > - Relation to prior work. Thank you for going through the relation to prior work. Some of your responses are a bit high-level. Can you be a bit more specific? For example, while the JMLR paper only establishes convergence of resolution invariance, can you comment on the convergence rate using your theory? And in the paragraph, "On Section 9.1 in the JMLR paper", can you be more specific, for example, can the new perspective in your theory give better results compared to those in Section 9?

---

> > > ### Author Response · Authors · 2023-08-19
> > > **Response to the reviewer**
> > >
> > > We would like to thank the reviewer for the helpful discussion. Your further question is addressed below. Let us know if you need more explanation.
> > >
> > > >Q10: For example, while the JMLR paper only establishes convergence of resolution invariance, can you comment on the convergence rate using your theory?
> > >
> > > **A10**: Our Theorem 3.3, in contrast to the JMLR paper, provides a more precise convergence rate along with its associated constants. Specifically, the super-resolution error depends on two components: the discretization error and the inherent super-resolution error of the continuous model itself. These correspond to the upper and lower terms on the right-hand side of Equation (5). For the former term, its convergence rate is determined by $e_{grid}(N_{grid})$. This rate is contingent upon the chosen integration format, and we furnish relevant discussions in lines 195-201 of the original paper. As an example, in the case of the integration used in FNO, the convergence rate is $O(1/N_{grid}^2)$.
> > >
> > > Note that: As the number of grid points, denoted as $N_{grid}$, approaches infinity, the super-resolution error diminishes towards zero. This is attributed to the fact that with an infinitely increasing grid point count, the model achieves predictions across all points and it becomes continuous so that there is no discretization error, resulting in a zero super-resolution error.
> > >
> > > The second term pertains to the super-resolution error, and its convergence rate is $O(1/N_{grid})$. Furthermore, we provide an accompanying constant for each of these convergence rates. These constants, concealed within the $O()$ notation, are influenced by the model parameter matrix norm and the Lipschitz continuity of the chosen basis and input functions. To encapsulate, our work presents a finer-grained convergence rate in comparison to the JMLR paper, along with the disclosure of the constants within these convergence rates.
> > >
> > > >Q11: And in the paragraph, "On Section 9.1 in the JMLR paper", can you be more specific, for example, can the new perspective in your theory give better results compared to those in Section 9?
> > >
> > > **A11**: Recall that we discuss section 9.1 in the JMLR paper since the reviewer asked that "how your perspective of neural
> > > operator compared to the set of neural operators used in their theory, defined in Section 9.1?"
> > >
> > > Here we provide more details.
> > >
> > > Equation (1) in our paper is essentially similar to the model $NO_n$ defined in the second equation on page 54 of the JMLR paper. They all adhere to the conventional paradigm of defining neural operators. Consequently, in our paper, Equation (1) falls under the category of preliminary groundwork. Our pivotal contribution resides in introducing the formulation of neural operators in the infinite-dimensional $l^2$ coefficient space, as outlined by Equation (2) in our work. Equation (2) in our paper constitutes the pivotal point of innovation. We ingeniously employ an orthogonal basis to transform the input functions into an equally infinite-dimensional vector space, albeit with real-valued elements. This strategic move enables the adoption of numerous tools from learning theory since the inputs are now an infinite-dimensional real number vector so that we can nontrivially extend traditional results on finite-dimensional real number vector learning. Conversely, if we were to adopt the conventional approach of defining models through function mappings, the analysis of such abstract infinite-dimensional functions becomes exceedingly challenging within a theoretical framework.
> > >
> > > >Q12: Detailed limitation?
> > >
> > > **A12**: Deep learning is often able to learn intricate models that exhibit the capability to generalize to unseen data, and this paradox has been a subject of significant interest. Dr. Albert Einstein has a famous quote: “In theory, theory and practice are the same. In practice, they are not.”
> > >
> > > Although we are confident that our theoretical framework offers superior guidance for comprehending the generalization, super-resolution, and discretization errors intrinsic to neural operators, there exist gaps between our theory and practice.
> > >
> > > For instance, in the generalization analysis of Theorem 3.1, the model parameter matrix norm could potentially be quite large in practical scenarios, which might lead to a comparatively loose bound. Therefore, the underlying assumption of our theorem is that when applying the same training procedures to two models, the obtained bound offers the predictive capability for the models' generalization performance. However, if entirely different training approaches are employed for the two models, there might exist a certain gap between theory and practice.

---

### Official Review · Reviewer_xMYc · 2023-07-24

**Soundness:** 3 good
**Presentation:** 2 fair
**Contribution:** 3 good
**Rating:** 5
**Confidence:** 4

**Summary:**

The paper attempts to provide theoretical analyses of Neural Operators (NOs) mainly considering the following aspects: 1. The generalization bound of NOs; 2. The discretization error of NOs; 3. The "super-resolution" (trained on low-resolution grid then evaluate on the high-resolution grid) error of NOs on the uniform grids. Several numerical experiments are carried out to validate the theory. The theory results also motivate the authors to come up with improved designs on existing NOs, including bases/integration schemes that suit specific PDE better, and generalization to unbounded domains

**Strengths:**

* A tighter and more general generalization bound compared to prior works;
* Advance in the understanding of discretization errors and super-resolution errors in NOs;
* The improved design of NOs is validated by extensive numerical experiments.


The paper in general has analyzed several crucial properties of NOs, which is timely, and its practical implication I believe will benefit the scientific ML community. Specifically given that the concept of doing super-resolution with NOs is often vaguely studied in many relevant works.

**Weaknesses:**

The overall presentation of the paper is fairly clear and easy to follow. However, some of the discussion in the experiment section is relatively vague, especially in section 6.3. As revealed by Theorem 3.3, the integration scheme along with basis selection has a crucial effect on the super-resolution error. The authors only briefly covered what basis and quadrature rule are used for the harmonic oscillator example, but then skim through the 3D DFT experiment, which makes it difficult to interpret the improvement in the results shown in Table 1. These details might be trivial for someone who is an expert in DFT, but they can help other audiences better understand the practical implication of the theorems.

**Questions:**

* One advantage of FNO is its superior computation efficiency on a uniform grid thanks to FFT. What is the rough estimate of the increase in the computation cost when using other integration schemes and bases?
* Following theorem 3.3, will the increase in truncated modes harm the super-resolution performance?

Possible typos:
* line 197, page 5: , $e_{grid}(N_{grid}) = 1/N^2_{grid}$ -> $e_{grid}(N_{grid}) = 1/N_{grid}$
* line 353, page 9: $u^2(x, t)$->$u(x, t)$

**Limitations:**

The paper did not discuss any limitation. While this is somewhat understandable as a theory paper, the authors can discuss the limitation of their theory results in terms of the application scope and its assumption.

---

> ### Author Rebuttal · Authors · 2023-08-08
>
> Thanks for your valuable comments. Your questions are addressed as follows.
>
> >Q1: Some of the discussion in the experiment section is relatively vague, especially in section 6.3. As revealed by Theorem 3.3, the integration scheme along with basis selection has a crucial effect on the super-resolution error. The authors only briefly covered what basis and quadrature rule are used for the harmonic oscillator example, but then skim through the 3D DFT experiment, which makes it difficult to interpret the improvement in the results shown in Table 1.
>
> **A1**: We provide more details for the numerical integration format and the orthogonal basis in the DFT experiments here and will add them to the revision.
>
> Quantum harmonic oscillator: The orthogonal basis is the  1D Hermite polynomial
> $
> \phi_i(x) = \dfrac{1}{\pi^{\frac{1}{4}} 2^{\frac{i}{2}}\sqrt{i!}}H_i(x)e^{-\frac{x^2}{2}}, x\in\mathbb{R},
> $
> The grid for training is the points in the Gauss-Hermite quadrature with 32 points, and the super-resolution testing grid is with degree 64.
>
> 3D DFT: Kohn-SHam-DFT offers natural basis sets. Additionally, grids with varying resolutions, characterized by levels, are provided based on the multi-center Gaussian nature of the functions in DFT. In our experiments, we select level 1 for training and level 2, which has more points, for testing.
> More concretely, for basis sets, common choices are the sto-3g basis where several primitive Gaussian orbitals are fitted to a single Slater-type orbital (STO), and 6-31g basis containing 6s6p Gaussian functions and 3d Polarization functions and 1d diffuse function. In the experiment, we used sto-3g basis sets, which can be orthogonalized using Gram-Schmidt orthogonalization.
> For sto-3g basis, the choice of quadrature points is related to the number of Gaussians used to construct each basis function. The process of selecting these quadrature points involves finding a set of points and weights that accurately approximate the integrals while minimizing computational cost.
>
> Generally, due to the distribution of atoms throughout the entire space, the electron density in a molecule (i.e., the input function of a molecule) resembles a mixture of Gaussian distribution centered around the different atoms within the molecule. As a result, quadrature points are selected densely around each atomic center to accommodate the multi-centered Gaussian nature of the molecular density function. Our sampling approach follows the interface provided by the commonly used quantum chemistry software, PySCF (https://pyscf.org/) and D4FT (ref 17 in the main paper). By providing the molecular formula and the desired grid density, PySCF returns the appropriate quadrature points to us.
>
> For the practical implication of the theorems by the experiments, we kindly refer the reviewer to lines 340-349 on page 9 of the main paper.
>
> >Q2: One advantage of FNO is its superior computation efficiency on a uniform grid thanks to FFT. What is the rough estimate of the increase in the computation cost when using other integration schemes and bases?
>
> **A2**: In FNO, the Fast Fourier Transform (FFT) is used to compute integrals, with a time complexity of $O(N_{\text{grid}}\log N_{\text{grid}})$. However, in practice, only the first  $N_{\text{modes}}$ integrations between the basis and input need to be calculated. This reduces the complexity to $O(N_{\text{modes}}N_{\text{grid}})$. Since $N_{\text{modes}} \ll N_{\text{grid}}$, even computing the numerical integration without FFT can ensure that the integration step does not become a bottleneck in the NO model's time complexity. The computational efficiency of other integration schemes and bases is still $O(N_{\text{modes}}N_{\text{grid}})$, which has negligible additional cost compared to FFT and may even be faster than FFT since $N_{\text{modes}} \ll N_{\text{grid}}$. In practice, the follow-up work of FNO by the same authors, the Geo-FNO (ref 19 in the main paper), adopts direct integration with $O(N_{\text{modes}}N_{\text{grid}})$ complexity and does not use FFT.
>
> In our computational approach, to perform kernel transformations, a total of $N_{\text{modes}}$ numerical integrations are required to obtain $N_{\text{modes}}$ coefficients. The complexity of each numerical integration is proportional to $N_{\text{grid}}$, resulting in an overall complexity of $O(N_{\text{modes}}N_{\text{grid}})$. It is important to note that the computation for each numerical integration involves calculating a weighted average of the integrand's values at $N_{\text{grid}}$ grid points.
>
> In contrast, in the context of FFT, they are obligated to compute all $N_{\text{grid}}$ integrals, which introduces considerable redundancy. Although their time complexity per integral reduces from $N_{\text{grid}}$ to $\log N_{\text{grid}}$, the need to compute all integrals adds significant overhead.
>
> >Q3: Following theorem 3.3, will the increase in truncated modes harm the super-resolution performance?
>
> **A3**: Yes, introducing more modes could potentially lead to an increase in the super-resolution error. This is because, within the bound of the super-resolution error, we can observe that the right-hand side includes the Lipschitz continuity constants of these orthogonal bases. If we incorporate larger modes, implying higher-frequency or higher-order basis functions, e.g., $\phi_k(x) = \sin(kx)$ with large $k$ value, they typically exhibit steeper profiles with larger Lipschitz constants. Consequently, this could result in less accurate numerical integrations on the training grid, hindering the generalization to more accurate testing grids. The lack of precision in numerical integration during training could consequently lead to significant discrepancies between predictions in the training and testing phases.
>
> >Q4: On the typos and limitations.
>
> **A4**: Thanks for pointing out our typo and not discussing limitations, we will carefully check all the content and add discussion in the revision.

---

> > ### Comment · Reviewer_xMYc · 2023-08-19
> > **Reply to the author**
> >
> > I would like to thank the authors for their detailed responses and the efforts they have made during the rebuttal period. Most of my concerns in the experiment section are addressed. In general, I think the theory results presented in the paper will be helpful for practitioners in the neural operator communities. The authors also further clarify the difference between their work and some of the existing works (in particular the JMLR paper) and what are the new implications. I remain positive towards the paper and thus keep my score unchanged.

---

### Author Rebuttal · Authors · 2023-08-08

We would like to thank all reviewers for their valuable comments. Here, we present general responses for understanding the manuscript better. Your specific questions are answered respectively.

>P1 (reviewer 59nv): On the contribution.

We present two primary theoretical contributions, we innovatively transform the generalization and super-resolution problems in the infinite function spaces into analyzable problems within the more tractable infinite-dimensional $l^2$ space. In this context, elements are represented as infinite-dimensional numerical vectors rather than abstract functions, facilitating a more comprehensive analysis. This transformation is inherently innovative.

Secondly, we investigate the generalization and super-resolution properties of neural operators in the $l^2$ space. The extension of finite-dimensional theories to infinite-dimensional domains is far from trivial. As a culmination of our efforts, we derive the generalized error, discretization error, and super-resolution error of the neural operator as an infinite-dimensional mapping. Notably, our work stands as the pioneering analysis conducted within an infinite-dimensional space.

>P2 (reviewer 2NvK): Intuitions of the theorems' proofs.

Theorem 3.1:
The proof of Theorem 3.1 primarily delves into the examination of the neural operator model's robustness and Lipschitz continuity. In Proposition 3.1, we establish that the activation functions in the coefficient space maintain their Lipschitz continuity as in the original space. Subsequently, the Lipschitz continuity of the linear mappings within the neural operator is determined by the matrix norms of these linear mappings. Combining these norms through multiplication yields the Lipschitz continuity of the entire model. By inserting these findings into the relationship formula encompassing robustness (provided by Ref 29 in the main paper), Lipschitz continuity, and generalization, the proof is concluded.

Theorem 3.2: The proof of Theorem 3.2 establishes the discretization error of the neural operator that error essentially originates from the numerical integration component within the neural operator, as in an ideal scenario, a continuous neural operator would utilize continuous integration rather than numerical methods. Thus, fundamentally, this proof aims to bound the disparity between numerical and continuous integration. Consequently, the ultimate outcome indicates that the discretization error is determined by the accuracy of numerical integration and the nature of the integrand. This integrand effectively represents the outputs of the model's intermediate states, and its complexity is expressed through the matrix norm of the model parameters.

Theorem 3.3: Firstly, we bound the prediction error of continuous NOs across all points in the domain $\Omega$ (i.e., the super-resolution error of continuous NOs). During the training phase, NOs are trained on a finite training grid, leading to this error. Subsequently, we bound the discrepancy between continuous and discrete NOs, corresponding to the discretization error stated in Theorem 3.2. These two terms are reflected in Theorem 3.3.

>P3 (reviewer 2NvK): Intuition of the 3 theorems.

Theorem 3.1: The generalization bounds, similar to vanilla neural nets, rely on the products of multilayer parameter norms. It aligns with the effectiveness of our commonly used L2 regularization. Generally, employing regularization tends to lead to smaller norms of model parameters, which in turn correspond to reduced generalization error. Additionally, our bound suggests that increasing the number of modes does not necessarily increase the complexity of the model.

On the right-hand side of Theorem 3.1, the second term contains $N_{train}$, indicating that as the training data increases, the generalization error diminishes. This aligns with our intuition. Furthermore, the factor $2K\log 2$ in the numerator signifies the vastness of the entire input data space, and our approach employs the concept of the covering number. Here, $K$ represents the so-called covering number, and a larger $K$ indicates that the input space is considerably expansive. Consequently, a greater amount of training data is required to effectively cover it, thereby yielding enhanced training outcomes. Conversely, if $K$ is smaller, it implies that the input space is relatively more compact, and thus, a reduced amount of data can still result in a well-performing model. In essence, $K$ intuitively reflects the efficiency of training data for a machine learning problem.

Theorem 3.2: The discretization error in discrete NOs relies on the norm of model parameters ($B_{k,i}, W_k$) and the accuracy of the integration method employed for integrating intermediate output functions (i.e., $e_{grid},e_{func}$). The dependence of discretization error on the accuracy of the integration scheme is inherently intuitive. This correlation arises from the fact that as the integration becomes more precise, the requirement for fewer grid points to achieve the same level of accuracy is observed. On the other hand, the norm of the model's parameters reflects the complexity of the integrated functions, which the model characterizes. A larger norm signifies heightened complexity in the functions represented by the model. This complexity implies that these functions are less easily approximated by values at finite points, leading to a potentially increased discretization error.

Theorem 3.3: The first term pertains to the integral error in Theorem 3.2, while the second term relates to the interpolation and generalization ability of NOs across the entire domain. In other words, the super-resolution error is essentially determined by two factors within the neural operator framework: the numerical error introduced by the employed numerical integration and the capability of its predictions to generalize beyond the training grid points.

---

### Author Response · Authors · 2023-08-21
**Looking forward to more discussions**

Dear Reviewers,

We extend our gratitude once again for your thorough and insightful feedback. We have responded to all reviewer comments to comprehensively address your concerns, and we are open to making further clarification and explanation.

With the conclusion of this rebuttal phase drawing near, we eagerly anticipate your continued input on our responses. We appreciate your active participation thus far, and should you have any remaining queries, please don't hesitate to reach out.

Warm regards,

The authors

---

### Decision · Program_Chairs · 2023-09-21

**Decision:**

Reject

**Comment:**

The reviewer raised concerns about the significance of theory in this paper and the relevance of the theory to their empirical study. After taking to the internal discussion, reviewers recommended for rejection.

We encourage the authors to consider the feedback provided in the review process.